# Validation of the Unesp-Botucatu pig composite acute pain scale (UPAPS) in piglets undergoing castration

I Robles[1], S. P. L. Luna[2], P. H. E. Trindade[2], M Lopez-Soriano[1], V. R. Merenda[1], A. V. Viscardi[3], E Tamminga[4], M. E. Lou[3], M. D. Pairis- Garcia[1]*

**1** Department of Population Health and Pathobiology, College of Veterinary Medicine, North Carolina State University, Raleigh, NC, United States of America, **2** Department of Veterinary Surgery and Animal Reproduction, School of Veterinary Medicine and Animal Science, São Paulo State University (Unesp), Botucatu, São Paulo, Brazil, **3** Department of Anatomy and Physiology, College of Veterinary Medicine, Kansas State University, Manhattan, KS, United States of America, **4** Prairie Swine Centre, Saskatoon, SK, Canada

* mpairis@ncsu.edu

**Data Availability Statement:** All relevant data are within the paper and its Supporting Information files.

## Abstract

To accurately assess pain and support broadly-based analgesic protocols to mitigate swine pain, it is imperative to develop and validate a species-specific pain scale. The objective of this study was to investigate the clinical validity and reliability of an acute pain scale (UPAPS) adapted for newborn piglets undergoing castration. Thirty-nine male piglets (five days of age, 1.62 ± 0.23 kg BW) served as their own control, were enrolled in the study and underwent castration in conjunction with an injectable analgesic administered one-hour post-castration (flunixin meglumine 2.2 mg/kg IM). An additional 10, non-painful female piglets were included to account for the effect of natural behavioral variation by day on pain scale results. Behavior of each piglet was video recorded continuously at four recording periods (24 h pre-castration, 15 min post-castration, 3 and 24 h post-castration). Pre- and post-operative pain was assessed by using a 4-point scale (score 0–3) including the following six behavioral items: posture, interaction and interest in surroundings, activity, attention to the affected area, nursing, and miscellaneous behavior. Behavior was assessed by two trained blinded observers and statistical analysis was performed using R software. Inter-observer agreement was very good (ICC = 0.81). The scale was unidimensional based on the principal component analysis, all items except for nursing were representative ($r_s \geq 0.74$) and had excellent internal consistency (Cronbach's alpha $\geq 0.85$). The sum of scores were higher in castrated piglets post-procedure compared to pre-procedure, and higher than in non-painful female piglets confirming responsiveness and construct validity, respectively. Scale sensitivity was good when piglets were awake (92.9%) and specificity was moderate (78.6%). The scale had excellent discriminatory ability (area under the curve > 0.92) and the optimal cut-off sum for analgesia was 4 out of 15. The UPAPS scale is a valid and reliable clinical tool to assess acute pain in castrated pre-weaned piglets.

**Funding:** MPG and AV received funding from the National Pork Board #19-065 (pork.org). The funder has no role in study design, data collection and analysis, decision to publish, or preparation of the manuscript.

**Competing interests:** The authors have declared that no competing interests exist.

## Introduction

In the United States (US), approximately 94 million pigs are castrated annually to prevent unwanted breeding [1], reduce aggression [2] and improve meat quality [3]. Despite the acknowledgment that castration is painful [4–7], and the economic benefits associated with pain relief and long-term weight gain in pigs [8], this procedure is not commonly performed in conjunction with pharmaceutical pain control in the US [9–11]. From a European perspective, many countries have passed legislation requiring mandatory pain relief for castrated piglets. However, no current studies to date have assessed actual implementation of pain relief on-farm, and older studies dating back to 2016 noted only 5% of male piglets in some European countries are castrated with pain relief [12]. Currently, the US has no Food and Drug Administration (FDA) approved drug specifically labelled for pain management in swine if veterinarians were mandated to provide pain control [1, 13]. In order to obtain FDA approval for the use of pain relief in pigs, a product must first demonstrate efficacy as supported through clinically validated end-points [1].

Pain associated with castration can be assessed using a variety of measures, including quantifying deviations of a piglet's typical behavioral repertoire [14]. Deviations in maintenance behaviors (e.g., sleeping, nursing, walking, drinking) as well as increases in pain-associated behaviors (e.g., tail wagging, rump scratching and prostration) have been most frequently used to quantify castration pain [15–17]. More recently, a grimace pain scale in castrated pigs has been developed and shows great potential as a future tool for evaluating piglet pain sensitivity [16, 18].

To accurately assess pain states and support broadly-based pain management protocols to mitigate castration pain, the development and validation of objective methods is the next logical step. Pain assessment scales for use in animals show promise given these tools are designed to directly quantify individual animal responses to painful stimuli, yield reliability and validity within and between observers [14]. When reviewing the pain literature in other species, pain assessment scales have been widely adopted as objective tools to quantify pain states in humans [19], domesticated dogs [20–22], cats [23–25], ruminants [26, 27], horses [28–30] and weaned pigs [31]. In addition, pain assessment scales offer advantages for use in livestock as these scales can be applied non-invasively on diverse farm settings, with little to no over-head cost required [14, 31].

In 2020, the Unesp-Botucatu pig composite acute pain scale (UPAPS) was developed specifically for weaned pigs (age: 38 ± 3 days) experiencing pain associated with castration [31]. The UPAPS has good to very good intra and inter-observer agreement, excellent predictive and concurrent criterion validity, and responsiveness [31]. Although this scale was validated as a reliable tool to quantify pain in pigs, castration performed on commercial farm systems is typically conducted on pre-weaned piglets (less than 7 days of age) housed directly with the sow and littermates. Consequently, clinical validation of this scale in a more appropriate age category and environmental setting is necessary as age and social dynamics influence the behavioral repertoire of pigs (i.e. pre-weaned pigs will spend more time nursing and sleeping than older pigs [32]). Therefore, the objective of this study was to investigate the reliability and validity of the UPAPS in assessing postoperative pain in pre-weaned piglets undergoing castration.

## Materials and methods

Animal use and procedures were approved by North Carolina State University Animal Care and Use Committee (Animal Utilization Protocol #19–796).

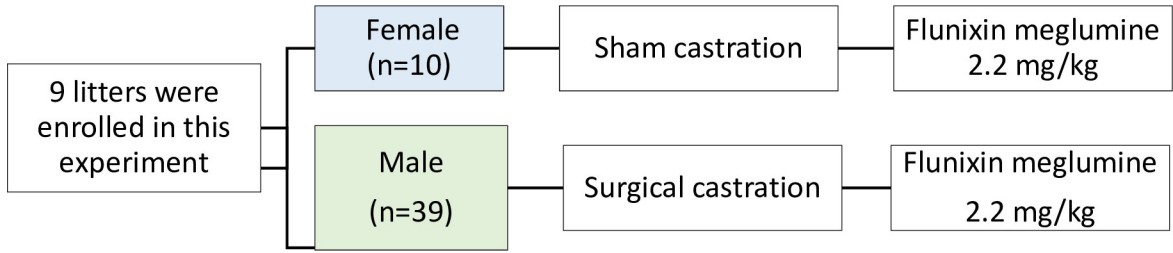

**Fig 1. Flow of the experimental design.**

## Study animals and housing

A total of 39 Yorkshire-Landrace x Duroc piglets from nine litters (five days of age, 1.62 ± 0.23 kg BW; Fig 1) were enrolled in the study. All male piglets in each litter were enrolled in each litter for the study (range: 1–10 males enrolled per litter), and, on average, two female piglets per litter were enrolled in addition. Given individual variation in pain sensitivity, the experiment was designed to allow male piglets to serve as their own control, thus assessing and comparing pain scores prior to and following a painful event. Male piglets were weighed and individually identified using a permanent marker on the back of the pig. The age category selected for this study was five days of age to simulate typical commercial production practices in the US [33–35].

Concurrently, 10 female piglets were enrolled simultaneously on the study and underwent a simulated castration (sham procedure) in which they were transported and handled in the same manner as males, but no incisions were made. Female piglets in non-pain states were enrolled to account for the effect of natural behavioral variation by day on pain scale results. Thus, female piglets were not considered true controls.

Sows and piglets were housed in individual farrowing crates (0.8 x 2.3 m) on fully slatted floors within one farrowing room at North Carolina State University (NCSU) Swine Educational Unit (Raleigh, North Carolina, US). The farrowing room was mechanically ventilated, and the temperature was maintained at 22˚ ± 1.0˚ C. Light was provided daily from 0630 to 1630 h. Creep area for piglets was heated to approximately 30–35˚C using heating lamps. Sows had *ad libitum* access to feed and water via awater nipple and stainless steel feeder (diet met or exceeded National Research Council [36] nutrient requirements for lactating sows) while housed in the farrowing system.

**Castration procedure.** One trained person from NCSU performed castration (IG) on all enrolled male piglets (n = 39) on the same day between 0800-1000h. Piglets were moved to the alleyway via a transport cart and individually suspended by the hind legs while the procedure was performed. Once the piglet was secure, two, 2.5 cm incisions were made into the scrotal skin over each testicle using a scalpel blade. Spermatic cords were torn, testicles removed, and iodine was sprayed over the incision site to reduce infection risk. General or local anesthesia was not administered at the time of castration and is standard practice in the US. Currently in the US, there are no FDA approved drugs specifically labeled to manage pain in swine. Therefore, the use of any analgesic/anesthetic protocol to mitigate castration pain must be administered under the direct supervision of a veterinarian in accordance with the Animal Medicinal Drug Use Clarification Act of 1994. Piglets enrolled on this study were castrated according to the current standard operating procedure (SOP) approved on farm by the attending veterinarian and the protocol did not include extra-label administration of an analgesic/anesthetic at castration. All male piglets located at this facility are required to undergo castration prior to weaning, therefore the castration procedure would have occurred regardless of the research project being conducted. It should be noted that although pain mitigation was not provided

pre-operatively to comply with farm SOP, all piglets enrolled in this research project received post-operative pain management as approved by the IACUC research protocol one-hour post-castration (2.2 mg/kg flunixin meglumine IM; Merck Animal Health, Millsboro, DE, US). Piglets enrolled on the study had no other procedure performed during the trial period (i.e., ear notching, injection, or tail docking).

## Behavioral data collection

Piglets were recorded using high-definition video cameras (AMCREST IP3M-943B; Houston, TX, US; one camera per crate) mounted 2.4 m above the ground and angled 45° above the farrowing crates. Behavior for each piglet was recorded continuously 1h in duration for the following four recording periods: M1 (24h pre-procedure), M2 (15 min post-procedure), M3 (3h post-procedure), M4 (24h post-procedure; Fig 2). At the end of the trial, one individual (MPG), not involved in the behavioral analysis, downloaded videos. For each video, four-minute video clips were trimmed from each recording period (M1-M4) to be evaluated by observers for pain scale validation. Video clips were obtained at the 16-min mark for each recording period (i.e., 16 min after the start of each recording period) until the 20-min mark. The 16-min mark was randomly selected to eliminate observer bias in potentially selecting a video segment with specific behavioral displays and frequencies. Once the continuous 4-min video clips were downloaded, clips were randomized and masked for identifying information (i.e., time stamp, date etc.) prior to initiation of pain scale validation. Observers assessed one piglet at a time for each observation. Potential bias may have occurred if observers were able to differentiate piglet sex. However, camera position and angle made it difficult to differentiate specific details on pigs and genitals were not easily discernable when reviewing videos prior to analysis, therefore, assumed bias level was minimum.

**Observer training.** The video clips from this study were analyzed by two female observers (IR, MEL). Prior to data collection initiation, the two observers underwent three training

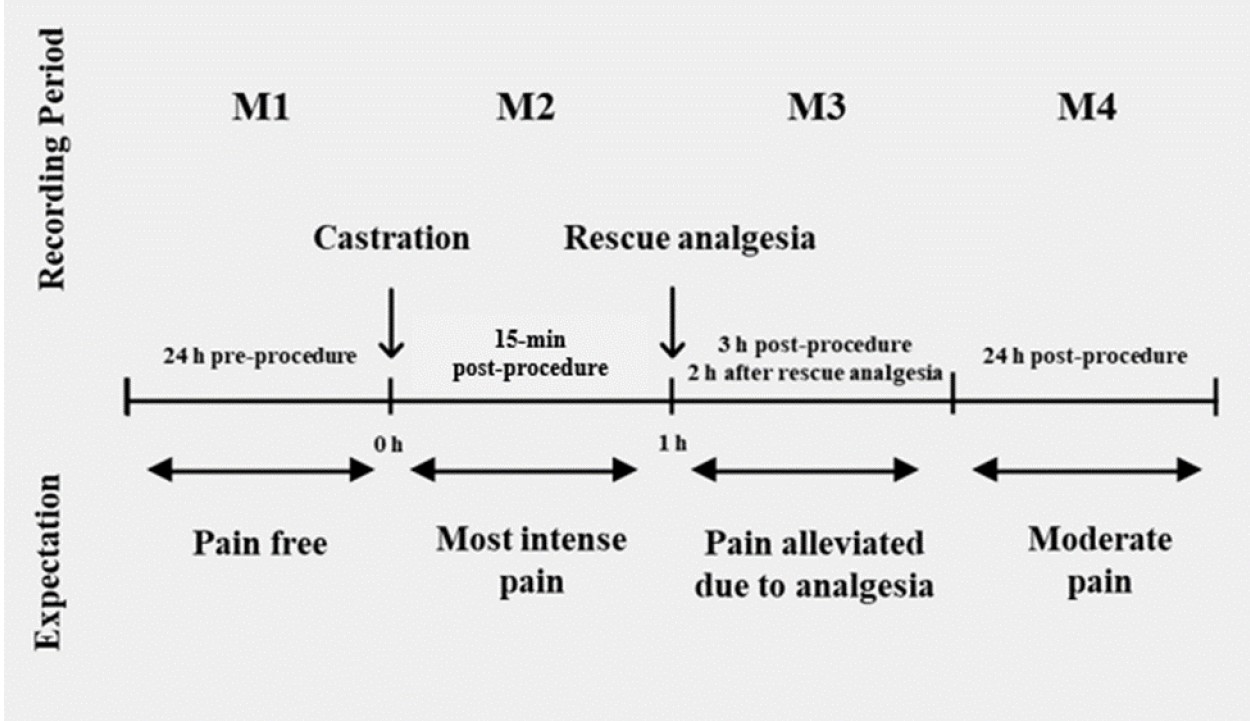

**Fig 2. Timeline of the periods of observations used for pain scale validation.**

sessions (2h/training session) where observers studied components of the scale, observed, and discussed examples of each behavioral score/criteria and practiced scoring for 90 min. Subsequently, inter-observer reliability was assessed, and intra-class correlation coefficient (ICC) was calculated by having each observer score twelve animals individually. Videos were observed in random order and observers were masked to day and time of day.

**Scale adaptations.** The pain scale validated in this study evaluated six behavioral items, with each item sub-categorized into four descriptive levels. A numerical score was designated from "0" to "3", with a "0" representing normal behavior (free of pain) and "3" corresponding to pronounced behavioral deviation (Table 1). For the purposes of this study, two adaptations

**Table 1. The Unesp-Botucatu composite pain scale for assessing postoperative pain in piglets (UPAPS).**

| Item | Score | Score/criterion | Links to videos |
|---|---|---|---|
| Posture | 0 | Normal[1] (any position, apparent comfort, relaxed muscles) or sleeping | https://youtu.be/ojvbFAgCWo0 |
| | 1 | Changes posture, with discomfort[2] | https://youtu.be/SpaWsFCrPxE |
| | 2 | Changes posture, with discomfort, and protects the affected area[3] | https://youtu.be/VjSlsRrG8yA |
| | 3 | Quiet, tense, and back arched[4] | https://youtu.be/gHUWWf5836M |
| Interaction and interest in the surroundings | 0 | Interacts with other animals; interested in the surroundings[5] or sleeping | https://youtu.be/6s8xEn3QR3c |
| | 1 | Only interacts if stimulated by other animals; interested in the surroundings. | https://youtu.be/9pMULbCO_WE |
| | 2 | Occasionally moves away from the other animals, but accepts approaches[6]; shows little interest in the surroundings | https://youtu.be/hp4wqdcHanA |
| | 3 | Moves or runs away from other animals and does not allow approaches; disinterested in the surroundings | https://youtu.be/se7ooYXcWFw |
| Activity | 0 | Moves normally[7] or sleeping | https://youtu.be/cC75t7L5-YA |
| | 1 | Moves with less frequency[8] | https://youtu.be/lQo9wq8LAn8 |
| | 2 | Moves constantly, restless[9] | https://youtu.be/YQRJjijLvpk |
| | 3 | Reluctant to move or does not move[10] | https://youtu.be/Zyx0G3Wpt8o |
| Nursing* | 0 | Actively nursing | https://youtu.be/6e2iYNKVFNY |
| | 1 | Not nursing when less than 50% of the piglets are nursing | https://youtu.be/3xJuMo2t7b4 |
| | 2 | Unsuccessful attempt at nursing (one or more attempts) | https://youtu.be/3PLgcf9ceQ8 |
| | 3 | Not nursing when 50% or more of the piglets are nursing | https://youtu.be/TZcB5PbsTT8 |
| Attention to the affected area | | A. Elevates pelvic limb or alternates the support of the pelvic limb | https://youtu.be/UD99ftO7HE0 |
| | | B. Scratches or rubs the painful area[11] | https://youtu.be/7idfFk1harE |
| | | C. Moves and/or runs away and/or jumps after injury of the affected area | https://youtu.be/u-Pqubom278 |
| | | D. Sits with difficulty | https://youtu.be/ETNEOCVV4h0 |
| | 0 | All the above behaviors are absent | |
| | 1 | Presence of one of the above behaviors | |
| | 2 | Presence of two of the above behaviors | |
| | 3 | Presence of three or all the above behaviors | |

(*Continued*)

**Table 1.** (Continued)

| Item | Score | Score/criterion | Links to videos |
|---|---|---|---|
| Miscellaneous behaviors | | A. Wags tail continuously and intensely[12] | https://youtu.be/pU5dGZFNRHc |
| | | B. Bites the bars or objects | https://youtu.be/cF3dsq7gMtk |
| | | C. The head is below the line of the spinal column. | https://youtu.be/ZcIgngclRpI |
| | | D. Presents difficulty in overcoming obstacles (example: other animal) | https://youtu.be/HlvdOI3lGuY |
| | 0 | All the above behaviors are absent | |
| | 1 | Presence of one of the above behaviors | |
| | 2 | Presence of two of the above behaviors | |
| | 3 | Presence of three or all the above behaviors | |

[1] Defined as any position in which the piglet demonstrates apparent comfort and relaxed muscles. Postures considered normal which demonstrate piglet comfort include but are not limited to piglets laying in lateral or sternal recumbency either separately or amongst the litter. Normal postural changes include a quick transition from standing to lying and lying to standing and typically result in the entire body descending onto or away from the floor at relatively the same time. Postural positions are most likely driven by external stimuli (e.g., littermate stands up or over piglets while lying). While standing, the piglet's head is parallel to the spine and the pig will frequently explore the environment by changing head positions (elevating head to explore surrounding area and placing head towards floor to explore ground).

[2] Defined as a piglet demonstrating a deviation from normal behavior in which muscles may be tensed, postural change transitions occur more slowly and may occur more frequently. Piglets demonstrating changing posture with discomfort may often extend the rear legs to elevate the rump off the ground while maintaining physical contact to the floor with the front legs and head. Transition from standing to lying or lying to standing also occurs more slowly in which the piglet may remain motionless for short time periods and change posture in the absence of external stimuli such as other littermates.

[3] Demonstrated behaviorally as a piglet performing a tucked tail posture in which the rear legs are slight flexed, and the piglet orients the castration site towards the walls or corner of the farrowing stall or runs away when another pig touches the affected area.

[4] Typically accompanied with little to no movement or activity. Back has a defined arch, head of the piglet is below the level of the spine and piglet spends majority of the time not moving or within a small physical space of the pen.

[5] Animal interaction includes snout to snout contact with littermates but may also include rooting or nosing body parts of littermates including the flank, neck, and rear of the littermate of interest. Interest in the surrounding environment includes actively exploring area (i.e., walking, running to various parts of the stall) and actively rooting with the snout or mouthing items within the stall including but not limited to heat mat, bars, and floor.

[6] Accepts approach is to reciprocate interaction with another littermate without running away from littermate.

[7] Defined as exhibiting forward motion with all four legs in contact with the floor typically at a walking pace. Normal movement examples for piglets include, but is not limited to, exploring the stall floor by walking to and from areas surrounding the sow (with and without the presence of other littermates), attempting to nurse sow by rooting at udder, and interacting with littermates on heat mat.

[8] Defined as the piglet still showing activity but is less likely to move for long periods or cover larger areas of the stall and may not attempt to explore a part of the stall in which no other littermates are present, may not participate in social interactions with littermates, or their behavior deviates from that of littermates.

[9] Piglets will likely demonstrate activity at a fast walk or running pace for the majority of the observation. Restless piglets may also demonstrate startle behaviors in which the piglet will quickly freeze (standing with head elevated but not moving) and then resume fast-paced movement.

[10] Piglets are likely to be in a lying position for the majority of the observation and demonstrate behavior similar to moribund animals. Piglets reluctant to move will not be easily stimulated by littermates and may isolate themselves from littermates and reside in the corner of the stall. Some piglets may show this behavior while demonstrating a posture score of 3 and will show very little to no reactivity of activity occurring in the pen. Postural changes may occur but will be less frequent in nature and will likely result in little to no physical distance changed from original posture location.

[11] Requires physical contact of the surgical site with the farrowing environment (i.e., floor, penning, mat) or littermate. Piglets often perform scratching behavior in which the rear leg is extended to the ear, but this is not considered for purposes of scoring this item as it does not involve the painful area.

[12] Tail wagging with intensity is described both by the force in which the tail moves and the frequency. Intense tail wagging will involve movement of the entire tail and movement will occur in short rapid bursts (3+ more tail wags consecutively). Tail wagging can occur while the piglet is standing, lying, or sitting.

* Nursing behavior was observed and analyzed in the study but was excluded from the final pain scale table as it did not meet any of the validation criteria (principal component analysis, specificity, internal consistency, item-total correlation and responsiveness).

to the pain scale were adopted and included: 1) replacement of feeding behavior with nursing behavior and 2) addition of the behavior sleeping into the 0 sub-categories for the items posture, interaction and interest in the surroundings, and activity. At the initiation of data observation, piglet state was evaluated and categorized as either awake or asleep. Piglet state was then included in the statistical model to facilitate the discrimination between a piglet sleeping (classified as normal) compared to an awake piglet expressing normal non-painful behavior.

## Statistical analyses

All statistical analyses were performed one person (PHET) in software R using the integrated development environment RStudio (Version 4.0.2 [2020-06-22], RStudio, Inc.). The functions and packages were presented in the format "function{package}" and α was considered 5% in all tests. A minimum sample size of 6 would be necessary based on the original UPAPS study [31] difference of total pain sums before and after castration based on alpha 0.01 and power of 0.9 (http://biomath.info/power/). Statistical procedures are presented in Table 2, and the scores of two observers were included. Litter was included as a random effect for all analyses. For some analyses, different stratifications of the database were used as specified in Table 2. To exclude the effect of sleeping in the scale validation performance, scores pertaining only to awake piglets were analyzed for some statistical tests (Table 2).

# Results

## Pain scale adaptations

The final UPAPS scale is presented in Table 1. The UPAPS+nursing (nursing scored from 0–3 scale described in Table 1) was not retained after analyses as it did not meet any of the validation criteria (principal component analysis, specificity, internal consistency, item-total correlation, and responsiveness described in Table 2. Results for each test completed from Table 2 utilizing the original UPAPS+nursing pain scale can be located in supporting information (S1, S2 Figs and S1–S7 Tables).

## Distribution of scores

In castrated piglets the distribution of scores "1", "2", and "3" for all items occurred more frequently at the M2 and M3 recording period compared to M1 and M4 (Fig 3). Scores for most items were proportionally greater post-castration (M2) when compared to after the rescue analgesia was administered (M3), except for the items "attention to the affected area" and "miscellaneous behaviors". Scores at M4 tended to return to pre-castration values presented at the M1 recording period (Fig 3). The behaviors included in the item "attention to the affected area" were performed more frequently post-castration (M2) compared to pre-castration (M1). The most frequent behavior displayed immediately post-castration (M2) for the item "attention to the affected area" was "sits with difficulty" and "wags tail continuously and intensely" for the item "miscellaneous behaviors" (Figs 3 and 4).

## Inter-observer reliability

Inter-observer agreement for all items of the pain scale ranged from moderate to very good (Table 3), except for nursing which ranged from reasonable to good. The agreement on total sum of UPAPS was very good (0.81; Table 3).

## Multiple association

The Horn's Parallel analysis [38] was used to determine the optimal number of dimensions for retention and indicated one dimension for the PCA, providing statistical rationale behind why

**Table 2. Statistical methods used for refinement ($^R$) and validation ($^V$) of the Unesp-Botucatu pig acute pain scale (UPAPS) in the piglets post-castration.**

| Type of analysis* | Description | Groups analyzed¥ | Statistical test |
|---|---|---|---|
| Internal consistency $^{RV}$ | The consistency (interrelation) of the scores of each item of the scale were estimated. The analysis was performed for all grouped moments. | All castrated piglets, and awake castrated piglets | Cronbach's alpha coefficient (α; "cronbach{psy}") and McDonald's omega coefficient (ω; "omega {psy}") were used. Cronbach's α interpretation: 0.60–0.64, minimally acceptable; 0.65–0.69 acceptable; 0.70–0.74 good; 0.75–0.80 very good; and > 0.80 excellent [37]. McDonald's omega interpretation: 0.70–0.84 is acceptable and > 0.85 is strong [38] |
| Distribution of scores $^V$ | 1-Piglets were categorized as awake "0", awake $\leq$ 30 seconds "1" or sleeping "2". 2-Percentage of the frequency distribution of the presence of the scores "0", "1", "2" and "3" of each item of the UPAPS at each moment and all moments grouped (MG). | All castrated and awake piglets for all moments grouped (MG = M1+M2 +M3+M4) | Descriptive analysis. |
| Inter-observer reliability $^{RV}$ | Reproducibility (agreement matrix)—the level of agreement between the two observers, using the scores for each item and the total sum of the scale was assessed. | All castrated piglets | For the scores of the items of the UPAPS, the weighted kappa coefficient ($k_w$) was used; the disagreements were weighted according to their distance to the square of perfect agreement ("cohen. kappa{psych}"). The 95% confidence interval (CI) $k_w$ based in 1,001 replications by the bootstrap method was estimated ("boot.ci{boot}"). For the sum of the UPAPS, the consistency type intraclass correlation coefficient (ICC; "icc{irr}") and its 95% CI based in 1,001 replications by the bootstrap method ("boot.ci{boot}") was used. Interpretation of $k_w$ and ICC: very good 0.81–1.0; good: 0.61–0.80; moderate: 0.41–0.60; reasonable: 0.21–0.4; poor < 0.2. The $k_w$ and ICC > 0.50 were used as a criterion to refine the scale [24, 39, 40]. |
| Multiple association $^{RV}$ | The multiple association of the scale items with each other was analyzed at all moments grouped using principal component analysis, to define the number of dimensions determined by different variables that establish the scale extension. | All castrated piglets, andawake castrated piglets. | Principal component analysis based on correlation matrix ("princomp{stats}") andget_pca_("var {factoextra}"). Horn's Parallel analysis was performed as the method to determine the optimal number of dimensions to be retained in the PCA [41]. Loading value greater than 0.50 or lesser than —0.50 were considered for significant association. For the biplot confidence ellipses were built with significant level of 95% showing the density of piglets' scores at each moment. |
| Item-total correlation $^{RV}$ | The correlation of each item of the scale, after excluding the evaluated item, was estimated to analyze homogeneity, the inflationary items, and the relevance of each item of the scale. Analysis was performed for all grouped moments. As nursing was not included in UPAPS, nursing was correlated with the total sum of the UPAPS without any exclusion. | All castrated piglets, and awake castrated piglets | Spearman rank correlation coefficient ($r_2$; "rcorr {Hmisc}"). Interpretation of item-total correlation [39]. Interpretation of correlation $r$: suitable values 0.3–0.7 |
| Specificity $^{RV}$ and sensitivity $^{RV}$ | The scores of each item of the scale at M1 (for specificity) and at M2 (for sensitivity) were transformed into dichotomous (score "0"—absence of pain expression behavior for a given item; scores "1", "2" and "3"— presence of pain). For the total score of the scale, the percentage of piglets that had score < 4 at M1 and $\geq$ 4 (cut-off point) at M2 was considered specificity and sensitivity, respectively. | All castrated piglets, andawake castrated piglets. | Specificity, sensitivity and its 95% CI were calculated according to the bootstrap method described below ("epi.tests{epiR}"). Interpretation: excellent 95–100%; good 85–94.9%; moderate 70–84.9%; not specific or not sensitive <70% [30]. |

*(Continued)*

**Table 2.** (Continued)

| Type of analysis* | Description | Groups analyzed¥ | Statistical test |
|---|---|---|---|
| Responsiveness and construct validity [RV] | Responsiveness—the scores of each item, and total sum of the scale were compared over time (M1 vs M2 vs M3 vs M4). Interpretation: differences in scores are expected to occur as follows: M2 > M4 ≥ M3 > M1. Construct validity was determined using four methods: 1. The three hypothesis tests: i) if the scale really measures pain, the score after surgery (M2) should be higher than the preoperative score (M1 < M2), ii) if analgesia was adequate the score should decrease after analgesia (M2 > M3), iii) and over time (M2 > M4). 2. Known-group validity. The Sham female pain-free piglets should have lower scores than castrated male piglets suffering pain at M2. 3. Internal relationships among items according to all criteria used in statistical analysis (multiple association, internal consistency and item-total correlation) 4. Relationships with the scores of other instruments, as described for criterion validity | All castrated piglets, Awake castrated piglets. Awake castrated piglets and Awake non-painful female piglets | The model residuals ("residuals{stats}") for all the dependent variables did not fit into Gaussian distribution according to the quantile-quantile and histograms graphs ("qqnorm{stats}" and "histogram {lattice}"), thus, generalized mixed linear models ("glmer{lme4}") were applied. For the dichotomous variables based on behaviors of attention to the affected area and miscellaneous behaviors, logistic regression analysis ("glm{stats}") was applied. Moments and observers were included as fixed effects and the piglets nested in their litters as random effect in all the models. Tukey test was used as a post hoc test ("lsmeans{lsmeans}" and "cld {multcomp}") [27]. |
| Optimum cut-off point [RV] | The data relative to the indication of rescue analgesia was used to determine the optimal cut-off point, that is, the minimum score suggestive of the need for analgesic rescue or intervention M1 free of pain (preoperative), and M2 pain (postoperative) were used to determine optimal cut-off point. | Awake castrated piglets | Cut-off point was based on the Youden index (YI = [Sensitivity + Specificity]– 1), which determines the highest sensitivity and specificity value concurrently from the Receiver Operating Characteristic (ROC) curve ("roc{pROC}" and "ci.sp{pROC}"), providing a graphic image of the relation between the "true positives" (sensitivity) and the "false positives" (specificity). The discriminatory capacity of the scale was determined by the area under the curve (AUC). AUC values above 0.90 represent high accuracy discriminatory capacity of the scale [42]. In addition, the 95% CI was calculated from the Youden index by replicating the original ROC curve 1,001 times according to the bootstrap method ("ci. coords{pROC}" and "ci.auc{pROC}"). |

*All analyses [37–42] included both adaptations of the scale: UPAPS Unesp Botucatu pig composite acute pain scale; UPAPS+nursing behavior.

¥ Total piglet numbers included: Male piglets (n = 39 total, n = 14 awake), Female non-painful piglets (n = 10 total, n = 6 awake)

M1 (24-h pre-procedure), M2 (15 min post-procedure), M3 (3-hr post-procedure), M4 (24-h post-procedure)

the pain scale applied to pre-weaned piglets was unidimensional. All pain scale items contained high loading values in the 1st dimension, explaining 63% of the variance of the entire data set (Figs 5 and 6; Table 4). In Figs 5 and 6, the ellipses corresponding to post-castration pain (M2 and M3) are in the right side of the figures. On the left side are the ellipses corresponding to no pain and mild pain (M1 and M4 respectively). Pain (M2 and M3) influence all items on the scale because their vectors are directed to these ellipses.

## Item-total correlation

Item-total correlation ranged from 0.62 to 0.71 for all male piglets, and from 0.75 to 0.83 for awake male piglets therefore, all items were relevant (Table 5).

## Specificity and sensitivity

When all castrated piglet scores were included in the analyses, the UPAPS items and total score had moderate to good specificity but no sensitivity (Table 6). When only awake piglets were considered, UPAPS total score and items had moderate to good specificity and good

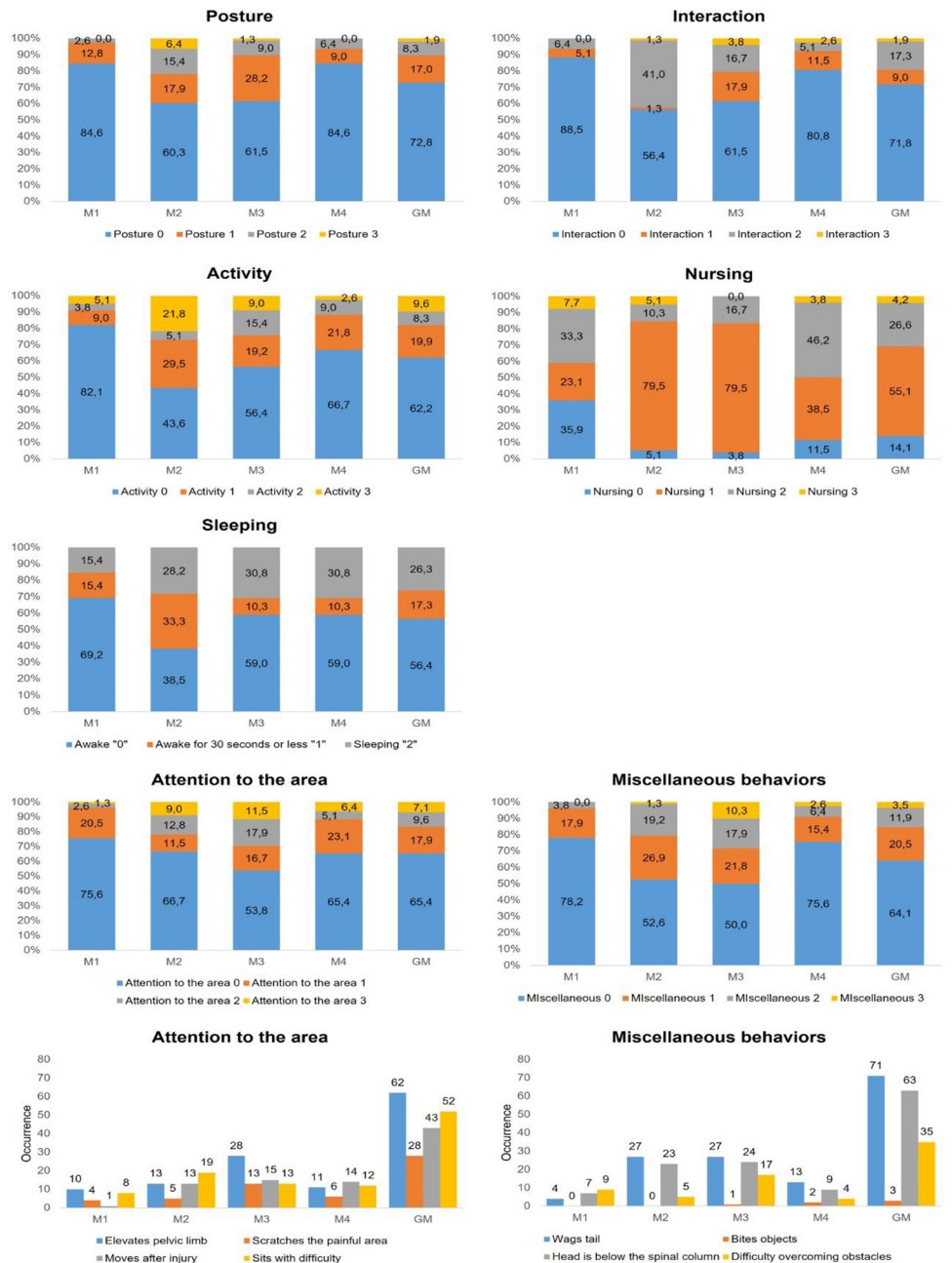

**Fig 3. Distribution of UPAPS items scores (percentage of occurrence) in castrated piglets (n = 39).** Legend: M1 (24h pre-procedure), M2 (15-min post-procedure, before rescue analgesia), M3 (3h post-procedure, after rescue analgesia), M4 (24h post-procedure) GM—data of the grouped moments (M1 + M2 + M3 + M4).

sensitivity, except for the following items: attention to the affected area (no specificity) and posture (no sensitivity; Table 6).

## Construct validity and responsiveness

The total sum of UPAPS were higher post-castration (M2 and M3) compared to pre-castration (M1) and 24h post-castration (M4). The total sum at M4 were intermediate between M1 and

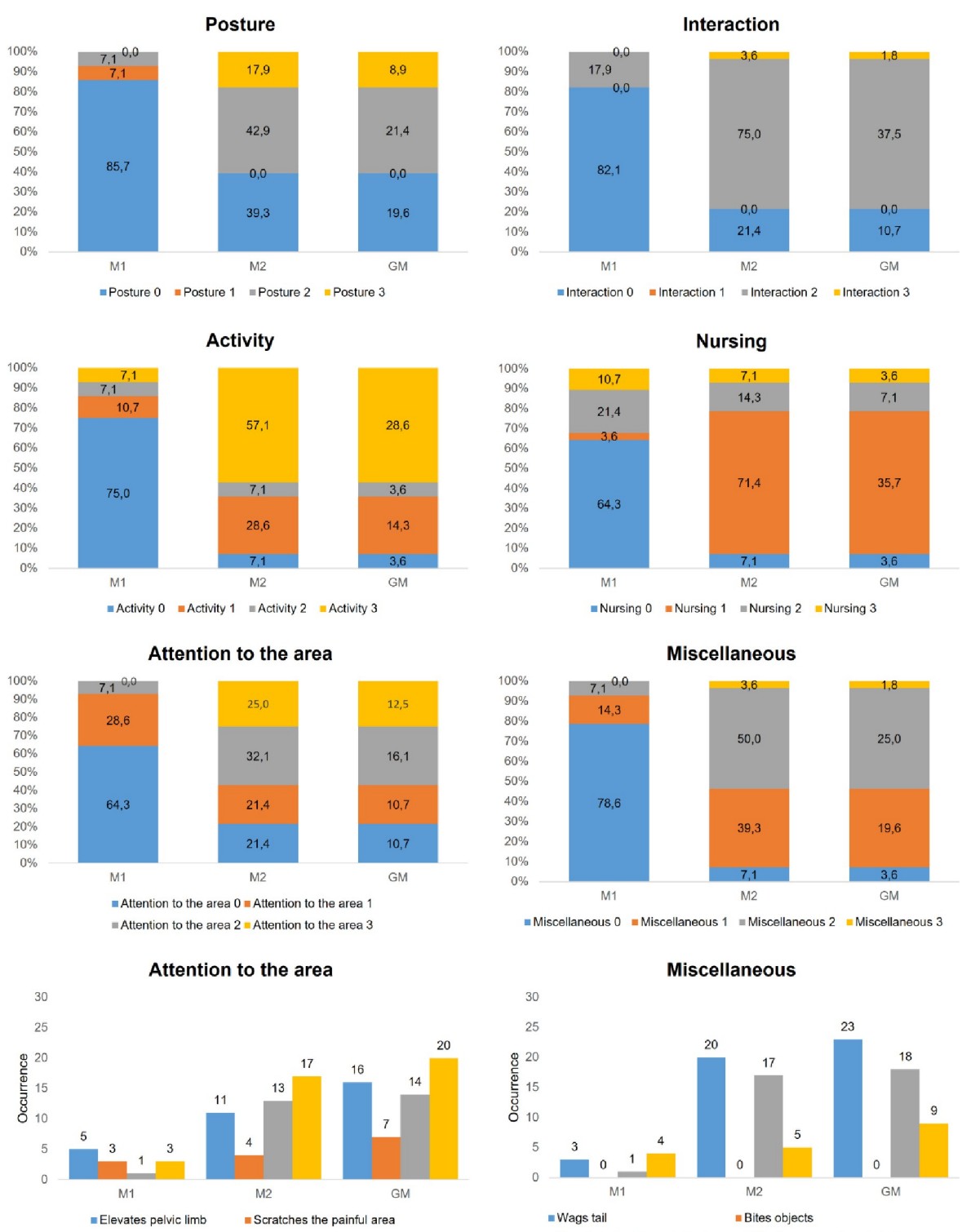

**Fig 4. Distribution of UPAPS items scores (percentage of occurrence) in awake castrated piglets (n = 14).** Legend: M1 (24h pre-procedure), M2 (15-min post-procedure, before rescue analgesia); GM—data of the grouped moments (M1 + M2).

**Table 3. Weighted kappa ($k_w$) or intra-class correlation coefficient (ICC) and confidence interval of the 95% from inter-observer matrix agreement of each item and the sum of the UPAPS applied for all castrated piglets (n = 39) over all moments.**

| Observer 1 vs Observer 2 | |
|---|---|
| **Items on the scale** | |
| **Items on the scale** | **$k_w$** |
| Posture | 0.74 (0.59–0.84) |
| Interaction and interest | 0.75 (0.60–0.85) |
| Activity | 0.68 (0.51–0.81) |
| Nursing | 0.57 (0.30–0.76) |
| Attention to the affected area | 0.70 (0.57–0.80) |
| Miscellaneous behaviors | 0.60 (0.45–0.72) |
| **Scales** | **ICC** |
| UPAPS | 0.81 (0.72–0.87) |

UPAPS—Unesp-Botucatu pig acute composite pain scale. $k_w$–weighted kappa coefficient; ICC—intraclass correlation coefficient; Interpretation of reliability: very good 0.81–1.0; good 0.61–0.80; moderate 0.41–0.60; reasonable 0.21–0.4; poor < 0.2.

M2/M3. Results demonstrated scale responsiveness to pain and time of day and distinguished moderate (M4) to intense pain (M2). However, the rescue analgesic was not effective, as demonstrated by the unresponsiveness of the scale to the rescue analgesia protocol (Fig 7A; Table 7).

Awake male piglets served as their own control and demonstrated higher scores after castration than before castration (Fig 7b; Table 8). After castration (M2), castrated piglets obtained higher scores than female non-painful piglets (Fig 8; Table 9), confirming construct validity and responsiveness.

### Internal consistency

The Cronbach's α coefficient of 0.85 and the McDonald's ω of 0.88 indicates excellent and strong internal consistency of UPAPS respectively (Table 9). Values of α = 0.89 and ω = 0.92 increased for castrated piglets awake (Table 10). The finding that the Cronbach's α of all individual items was below 0.85 (all piglets) or 0.89 (piglets awake), and the McDonald's ω of all individual items was below 0.88 (all piglets) or 0.92 (piglets awake) when each item was excluded for calculation, indicate that all items are relevant and contribute to the total score of UPAPS.

### Optimal rescue analgesia cut-off point

The optimal cut-off for UPAPS in this study was ≥ 4 and AUC was 0.92 demonstrating excellent discriminatory capacity of the instrument (Table 11; Fig 9).

### Discussion

Mitigating castration pain for piglets is a critical welfare issue and pain negatively impacts the affective state of the pig and productivity for the producer. Currently, the US has no FDA approved drug specifically labeled for pain management in swine [1, 13] and product approval can only be carried out by demonstrating drug efficacy as supported through clinically validated endpoints, such as behavior scales. In 2020, Luna and colleagues [31] developed and validated an acute pain scale to assess pain in castrated pigs post-weaning (age: 38 ± 3 days).

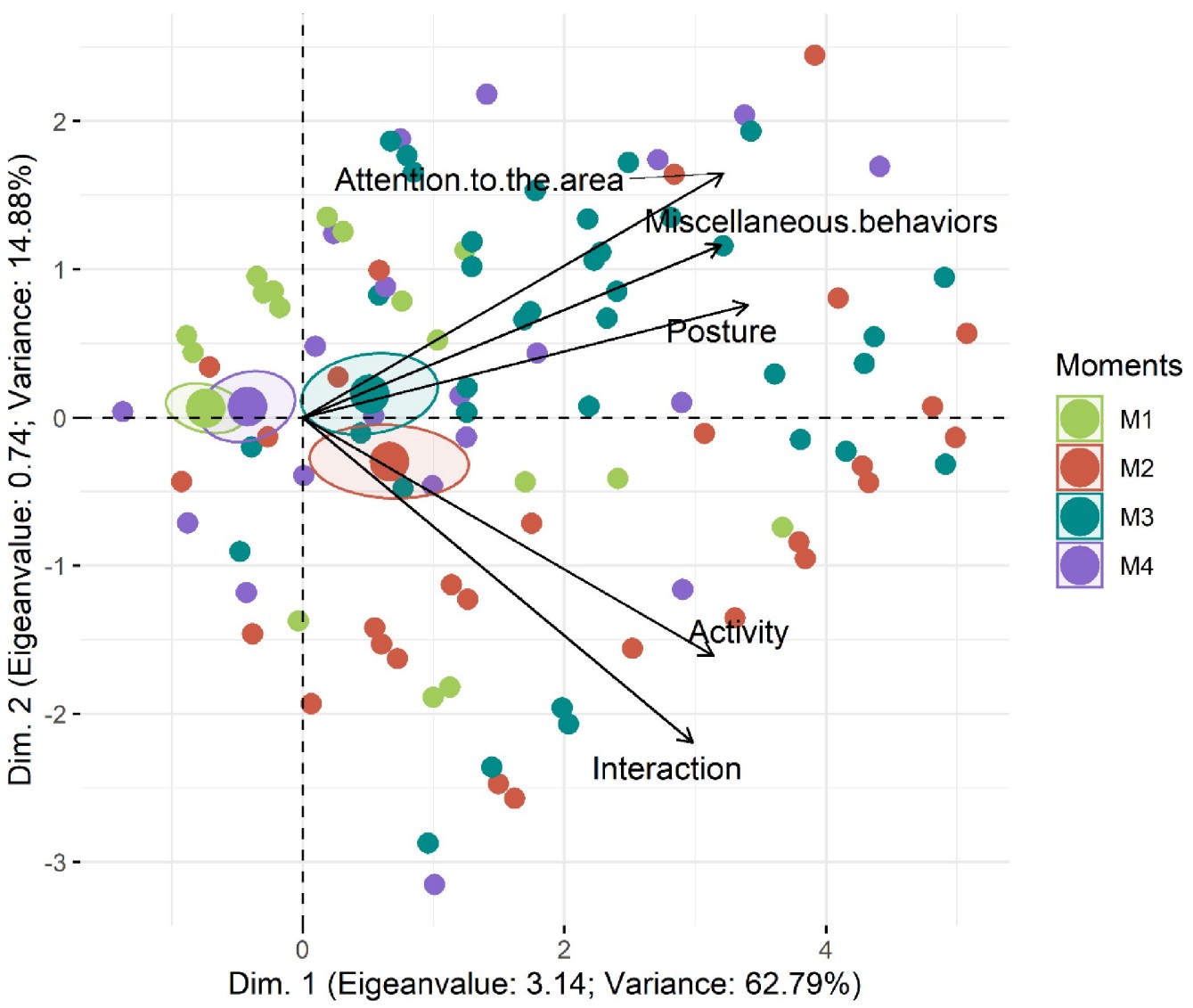

**Fig 5. Biplot of the principal component analysis of the UPAPS items applied to all castrated piglets (n = 39).** UPAPS–Unesp-Botucatu pig acute composite pain scale. Confidence ellipses indicate perioperative moments and pain scores. Moments: M1 (24h pre-procedure), M2 (15-min post-procedure, before rescue analgesia), M3 (3h post-procedure, after rescue analgesia), M4 (24h post-procedure). Ellipses were constructed according to the assessment. Each circle corresponds to the score from one observer attributed to each piglet at each moment.

Albeit promising, this tool required adaptation for use with pre-weaned piglets, since castration is performed within the first week of life on commercial farm systems and piglets are group housed with the sow and littermates. Thus, the objective of this study was to assess the reliability and validity of the UPAPS in assessing postoperative pain in pre-weaned piglets undergoing castration.

This study utilized the UPAPS scale originally developed for weaned pigs with the adaptations of two behaviors (sleep and nursing). Analysis of the adapted scale (UPAPS+nursing) revealed that nursing behavior as an item lacked internal consistency, was not homogenous and did not interrelate well with other items. When assessed independently, nursing behavior did not change significantly for any of the moments pre- and post-castration and between castrated piglets and non-painful female piglets, indicating a lack of responsiveness and

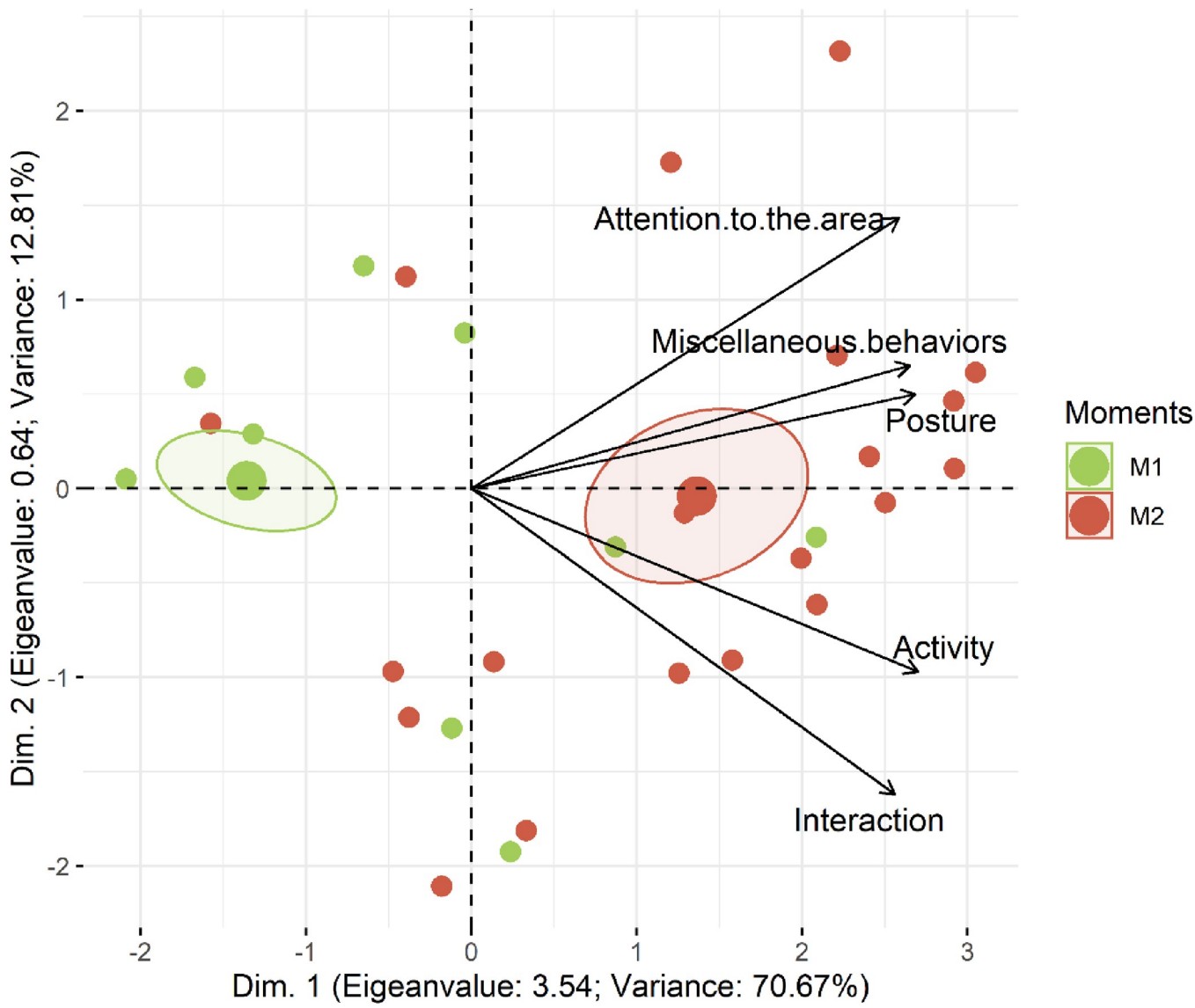

**Fig 6. Biplot of the principal component analysis of the UPAPS items applied to awake piglets (n = 14 castrated piglets).** UPAPS–Unesp-Botucatu pig acute composite pain scale. Confidence ellipses indicate perioperative moments and pain scores. Moments: M1 (24h pre-procedure), M2 (15-min post-procedure, before rescue analgesia). Ellipses were constructed according to the assessment. Each circle corresponds to the score from one observer attributed to each piglet at each moment.

specificity. These results are in contrast with previous studies utilizing nursing as an indicator of painful states [14]. Nursing behavior is often difficult to capture during short observation periods, as bouts are short in duration and social dynamics influence the length and frequency of nursing. Future use of nursing behavior as an indicator of pain state may be more effective if observation periods are longer and evaluated more frequently, therefore, the remaining discussion of this work will be based on the UPAPS excluding nursing behavior.

The UPAPS scale adapted for pre-weaned piglets has been validated as an effective tool to assess pain in castrated piglets utilizing the required validation assessments [23, 27, 43, 44]. This scale, as well as the UPAPS for pigs [31] and cattle [26], were considered unidimensional and all items in the scale had individual relevance and presented homogeneity. When

**Table 4. Loading values, eigenvalues and variance of the UPAPS items applied to castrated piglets (n = 39) and castrated piglets awake (n = 14) based on principal components analysis.**

| | Loading Values | | | |
|---|---|---|---|---|
| | UPAPS | | UPAPS (awake) | |
| Items on the scale | Dim.1 | Dim.2 | Dim.1 | Dim.2 |
| Posture | **0.84** | 0.19 | **0.86** | 0.16 |
| Interaction | **0.74** | **-0.55** | **0.82** | **-0.52** |
| Activity | **0.78** | -0.40 | **0.86** | -0.31 |
| Attention to the affected area | **0.80** | 0.41 | **0.82** | 0.46 |
| Miscellaneous behaviors | **0.79** | 0.29 | **0.84** | 0.21 |
| Eigen value | **3.14** | 0.74 | 3.54 | 0.64 |
| Variance | 62.78 | 14.88 | 70.67 | 12.80 |

UPAPS–Unesp-Botucatu pig acute composite pain scale. The structure was determined considering items with a loading value greater than 0.50 or lesser than-0.50 (in bold) and the only first dimension retained by Horn's Parallel analysis [38].

compared to the UPAPS scale for pigs, UPAPS for pre-weaned piglets had less specificity but better sensitivity with the majority of items achieving over 92%.

The UPAPS scale demonstrated behavioral deviations indicative of pain when comparing male piglet behavior pre-castration (M1) to post-castration (M2). Given individual variation in pain sensitivity, male piglets served as their own control by comparing pain scale scores prior to castration and at three moments post-castration, selected specifically to capture varying pain intensity levels. Post-castrated piglets demonstrated greater pain scores for all scale items (posture, interaction and interest, activity, attention to the affected area and miscellaneous) at M2 compared to M1. Behavioral changes post-castration included postural changes, decreased activity and interaction with surrounding environments as well as increased attention to affected area, sitting with difficulty, tail wagging and prostration (e.g. head below the spinal column). These results agree with previous work conducted using both pain scales and traditional behavioral observation with continuous recording. Piglets post-castration will more frequently display tense and stiff postures [14, 16, 17], decreased activity [45–47], and increased sleeping behavior [31].

In contrast to previous work, the UPAPS scale did not demonstrate changes in elevation of pelvic limb and scratching behavior when comparing M1 and M2 periods and is in agreement with Viscardi and Turner's work in 2018 [16]. Scratching behavior has been commonly used in previous castration studies to quantify local sensitization of the affected area, and has often been anecdotally associated with the wound healing process with some studies reporting

**Table 5. Item-total correlation of the UPAPS applied for all castrated piglets (n = 39).**

| Excluding each item below | Item-total (Spearman) | |
|---|---|---|
| | All castrated piglets | Castrated piglets awake |
| Posture | 0.68 | 0.77 |
| Interaction and interest | 0.62 | 0.74 |
| Activity | 0.71 | 0.83 |
| Attention to the affected area | 0.65 | 0.75 |
| Miscellaneous behaviors | 0.64 | 0.76 |

UPAPS: Unesp-Botucatu pig acute composite pain scale. Interpretation of Spearman's rank correlation coefficient ($r$): 0.3–0.7 = acceptable values [39].

**Table 6. Specificity and sensitivity and 95% confidence interval (CI) of the UPAPS applied for all castrated piglets (n = 39) and awake castrated piglets (n = 14).**

| | Specificity (%) at M1 | | | Sensitivity (%) at M2 | | |
|---|---|---|---|---|---|---|
| | **All castrated piglets** | | | | | |
| **Items** | **Estimated** | **CI** | | **Estimated** | **CI** | |
| Posture | **84.62** | 74.67 | 91.79 | 39.74 | 28.83 | 51.46 |
| Interaction and interest | **88.46** | 79.22 | 94.59 | 43.59 | 32.39 | 55.30 |
| Activity | **82.05** | 71.72 | 89.83 | 56.41 | 44.70 | 67.61 |
| Attention to the affected area | **75.64** | 64.60 | 84.65 | 33.33 | 23.06 | 44.92 |
| Miscellaneous behaviors | **78.21** | 67.41 | 86.76 | 47.44 | 36.01 | 59.07 |
| UPAPS (≥4) | **85.90** | 76.17 | 92.74 | 46.15 | 34.79 | 57.82 |
| | **Only awake castrated piglets** | | | | | |
| Posture | **85.71** | 67.33 | 95.97 | 60.71 | 40.58 | 78.50 |
| Interaction and interest | **82.14** | 63.11 | 93.94 | **78.57** | 59.05 | 91.70 |
| Activity | **75.00** | 55.13 | 89.31 | **92.86** | 76.50 | 99.12 |
| Attention to the affected area | 64.29 | 44.07 | 81.36 | **92.86** | 76.50 | 99.12 |
| Miscellaneous behaviors | **78.57** | 59.05 | 91.70 | **92.86** | 76.50 | 99.12 |
| UPAPS (≥ 4) | **78.57** | 59.05 | 91.70 | **92.86** | 76.50 | 99.12 |

UPAPS—Unesp-Botucatu pig acute composite pain scale. Interpretation of specificity and sensitivity: excellent 95–100%; good 85–94.9%; moderate 70–84.9%; not specific or sensitive < 70%; bold values ≥ 70% [30]. Specificity and sensitivity for the total scores were considered as the relation between the number of piglets that had pain scores < 4 at M1 and ≥ 4 at M2 and the total number of piglets, respectively. M1 (24-h pre-procedure), M2 (15 min post-procedure, before rescue analgesia)

scratching behavior to begin within the first hours post castration and continuing up to 4 days post castration [48]. However, the lack of change in scratching behavior in this study may be due to the fact that scratching behavior is performed infrequently (3.56 ± 0.83 bouts; [15]), in clusters, and in short-duration. Given this study only evaluated behavior for four consecutive minutes per moment, capturing pelvic limb alterations and scratching frequency may be difficult to achieve. Nevertheless, the remaining behaviors categorized within "attention to affected area" (moves/runs away, sits with difficulty) were different when comparing painful and non-painful states. Given the presence of any of the behaviors within the category were used for assessment, pain sensitivity can still be accurately assessed.

The internal consistency of this scale was excellent and comparable to pig [31], cattle [26] and cat [43, 44] scales. The scale demonstrated construct validity when comparing total scale score in castrated male piglets compared to non-painful female piglets. Non-painful female piglets were included in the study to assess the effect of sex, handling, time, and social dynamics on the pain scale during a non-pain state and were not considered true controls. In addition to construct validity, the UPAPS scale was responsive to pain state, time of day, and distinguished moderate to intense pain. The optimal cut-off point calling for rescue analgesia was the same as reported in the original UPAPS study [31] demonstrating excellent reproducibility for analgesic treatment decision-making using this tool. However, when assessing responsiveness to rescue analgesic protocol, the scale was not responsive and supports previous work demonstrating that a single administration of an NSAID post-castration does not effectively mitigate castration pain [14].

Although continuous behavioral assessment is currently considered the gold standard for pain evaluation [49], the UPAPS scale fulfilled a similar task within a short assessment time per piglet (only 4 minutes per period). Thus, this is a practical clinical tool that can be used in a veterinary hospital and/or on-farm. It is important to acknowledge that, although results are promising, the UPAPS pain scale should be further investigated as validation is a continuous process and should not be considered the one and only standard for pain assessment in pigs

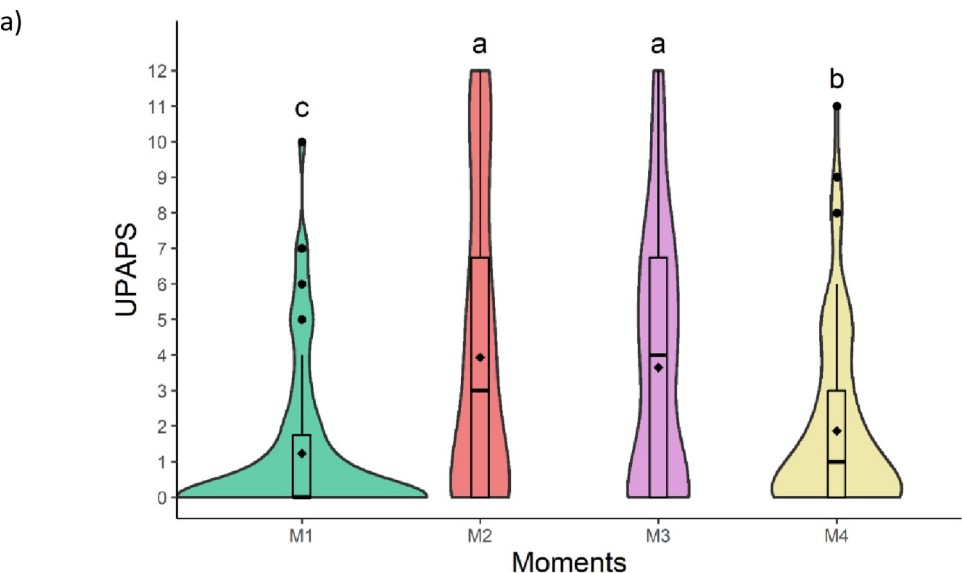

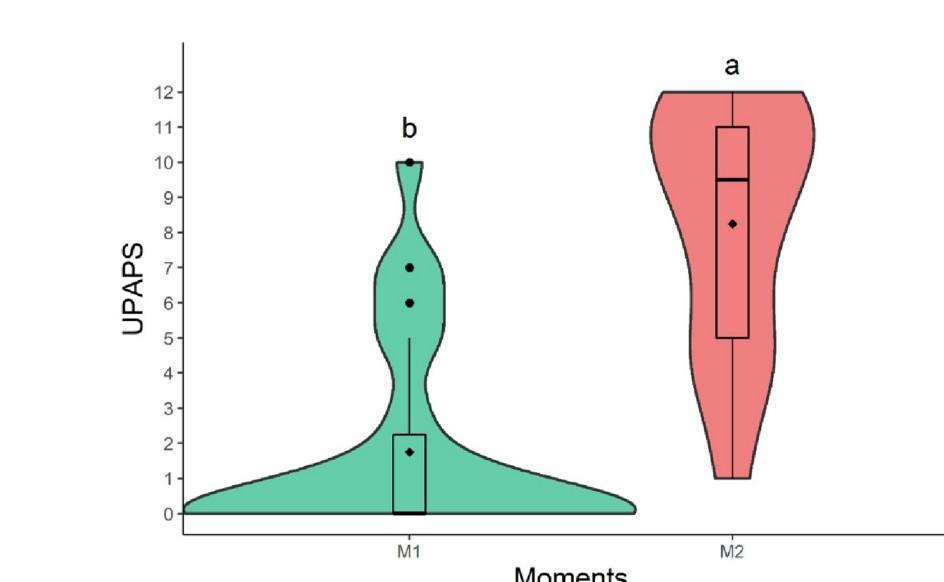

**Fig 7. Violin plot of the total sum (median/amplitude) of the a) UPAPS (all castrated piglets; n = 39) and b) UPAPS (awake castrated piglets; n = 14) before and after castration in piglets.** The violin contour represents the dispersion of data density, with the wider contour representing greater data density. The top and bottom box lines represent the interquartile range (25 to 75%), the line within the box represents the median, the extremes of the whiskers represent the minimum and maximum values, black lozenges (♦) represent the mean, black circles (•) represent outliers. UPAPS: Unesp-Botucatu pig acute composite pain scale. Different letters express significant differences between moments where a > b > c, according to the generalized mixed linear models [27]. M1 (24h pre-procedure), M2 (15-min post-procedure, before rescue analgesia), M3 (3h post-procedure, after rescue analgesia), M4 (24h post-procedure).

for future work, given several limitations associated with the method and the manner in which behavior was collected may have influenced the results. Piglet pain behaviors are often performed infrequently, subtly and can be significantly influenced by external factors such as the presence of humans, facility design and social syncing (isolation and desynchronization) [50].

**Table 7. Median (amplitude) of the UPAPS scores before and after castration in all castrated piglets (n = 39).**

| Behavioral pain scale items | M1 | M2 | M3 | M4 |
|---|---|---|---|---|
| **Posture** | 0[b] (0–2) | 0[a] (0–3) | 0[a] (0–3) | 0[b] (0–2) |
| **Interaction and Interest** | 0[d] (0–2) | 0[a] (0–3) | 0[b] (0–3) | 0[c] (0–3) |
| **Activity** | 0[c] (0–3) | 1[a] (0–3) | 0[ab] (0–3) | 0[bc] (0–3) |
| **Attention to the affected area** | 0[d] (0–3) | 0[b] (0–3) | 0[a] (0–3) | 0[c] (0–3) |
| Elevates pelvic limb | 0[b] (0–1) | 0[b] (0–1) | 0[a] (0–1) | 0[b] (0–1) |
| Scratches the painful area | 0 (0–1) | 0 (0–1) | 0 (0–1) | 0 (0–1) |
| Moves after injury | 0[b] (0–1) | 0[a] (0–1) | 0[a] (0–1) | 0[a] (0–1) |
| Sits with difficulty | 0 (0–1) | 0 (0–1) | 0 (0–1) | 0 (0–1) |
| **Miscellaneous behavior** | 0[b] (0–2) | 0[a] (0–3) | 0.5[a] (0–3) | 0[b] (0–3) |
| Wags tail | 0[b] (0–1) | 0[a] (0–1) | 0[a] (0–1) | 0[b] (0–1) |
| Bites objects | 0 (0–0) | 0 (0–0) | 0 (0–1) | 0 (0–1) |
| Head is below the spinal column | 0[b] (0–1) | 0[a] (0–1) | 0[a] (0–1) | 0[b] (0–1) |
| Difficulty overcoming obstacles | 0[ab] (0–1) | 0[b] (0–1) | 0[a] (0–1) | 0[b] (0–1) |
| **UPAPS** | 0[c] (0–10) | 3[a] (0–12) | 4[a] (0–12) | 1[b] (0–11) |

UPAPS—Unesp-Botucatu pig acute composite pain scale. Different letters express significant differences between moments where a > b > c > d, according to the mixed linear model [27]. M1 (24h pre-procedure), M2 (15-min post-procedure, before rescue analgesia), M3 (3h post-procedure, after rescue analgesia), M4 (24h post-procedure)

For example, work in horses [51], rabbits [52] and cats [53] demonstrated that human presence can influence pain behavior expressed by animals. In addition to the external factor of human presence, social dynamics within the litter can also influence behavior. Male piglet number per litter in this study varied, and all available male piglets were enrolled. Given this variation, the experiment was not designed to control for social cohesion [50] and future considerations should evaluate the pain scale using controlled pig number by litter and control for

**Table 8. Median (amplitude) of the UPAPS scores before and after castration in castrated piglets awake (n = 14) during the assessment.**

| Items | M1 | M2 |
|---|---|---|
| **Posture** | 0[b] (0–2) | 2[a] (0–3) |
| **Interaction and Interest** | 0[b] (0–2) | 2[a] (0–3) |
| **Activity** | 0[b] (0–3) | 3[a] (0–3) |
| **Attention to the affected area** | 0[b] (0–2) | 2[a] (0–3) |
| Elevates pelvic limb | 0[b] (0–1) | 0[a] (0–1) |
| Scratches the painful area | 0 (0–1) | 0 (0–1) |
| Moves after injury | 0[b] (0–1) | 0[a] (0–1) |
| Sits with difficulty | 0[b] (0–1) | 1[a] (0–1) |
| **Miscellaneous behavior** | 0[b] (0–2) | 2[a] (0–3) |
| Wags tail | 0[b] (0–1) | 1[a] (0–1) |
| Bites objects | 0 (0–0) | 0 (0–0) |
| Head is below the spinal column | 0[b] (0–1) | 1[a] (0–1) |
| Difficulty overcoming obstacles | 0 (0–1) | 0 (0–1) |
| **UPAPS** | 0[b] (0–10) | 9.5[a] (1–12) |

UPAPS—Unesp-Botucatu pig acute composite pain scale. Different letters express significant differences between moments where a > b, according to the mixed linear model [27]. M1: 24h pre-castration; M2: 15-min post-castration.

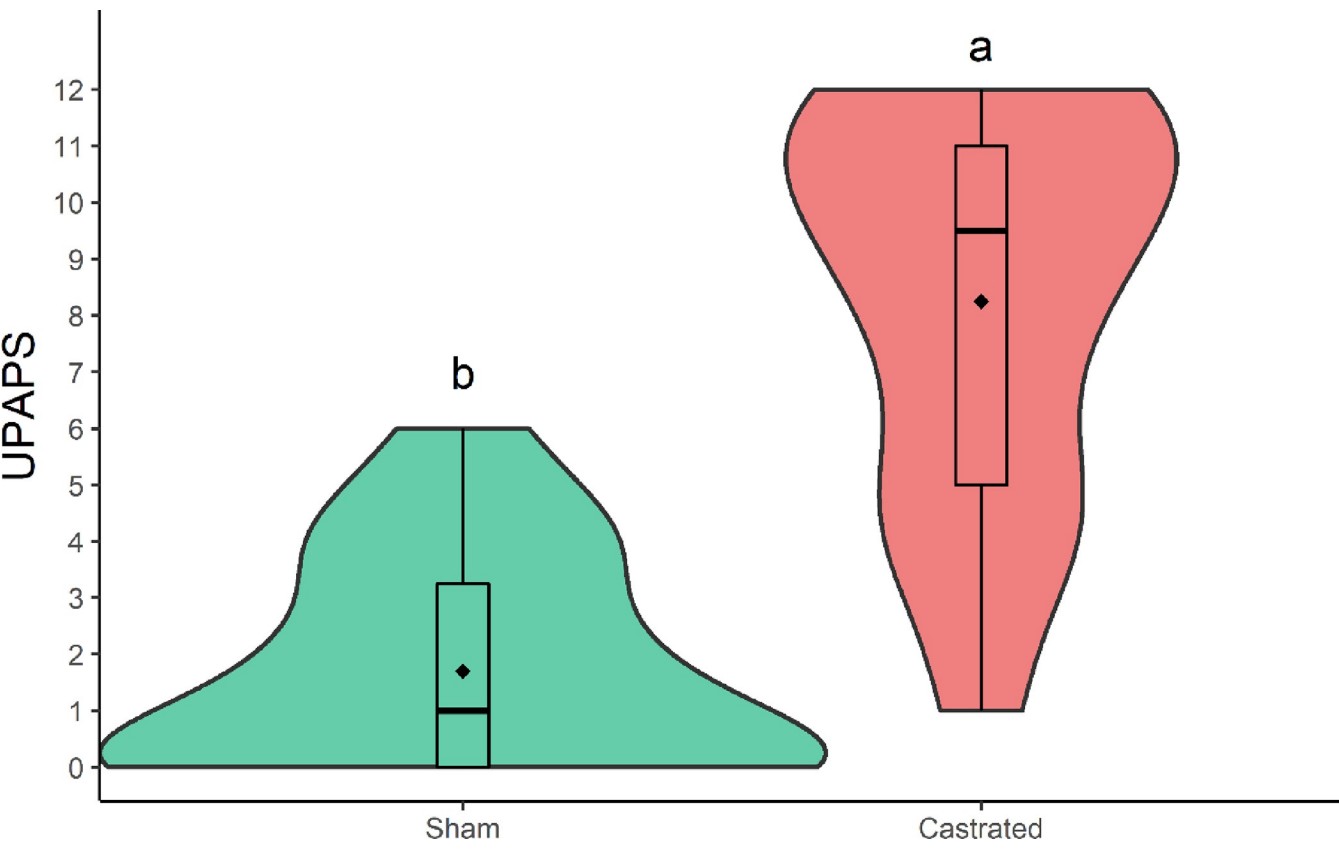

**Fig 8. Violin plot of the total sum (median/amplitude) of the a) UPAPS for castrated piglets (n = 14) and non-painful female piglets (n = 6) after castration (M2) and before rescue analgesia in piglets awake during the assessment.** The violin contour represents the dispersion of data density, with the wider contour representing greater data density. The top and bottom box lines represent the interquartile range (25 to 75%), the line within the box represents the median, the extremes of the whiskers represent the minimum and maximum values, black lozenges (♦) represent the mean, black circles (•) represent outliers. Sub-UPAPS: Unesp-Botucatu pig acute composite pain scale. Different letters express significant differences between moments where a > b > c, according to the generalized mixed linear models [27]. M2 (15-min post-procedure, before rescue analgesia).

an equal male: female ratio. Another possible investigation is to use machine learning algorithms to provide statistical weightings and define the importance of each behavior like performed in sheep [54].

Future work should not only address these external factors in the experimental design but should also include direct comparison of pain scale data to other validated measures to assess physiological pain response and pain behaviors using continuous sampling behavioral methodologies. For example, previous swine castration research has identified important pain behaviors specific to swine castration and include, prostrate, tail wag and scratching (summarized in [50]). For this study, these pain specific behaviors were categorized within the items "miscellaneous' and "attention to affected area' and scores were based on the cumulative presence/absence of each behavior. This approach did not allow data to be evaluated across behaviors but by a categorical score, thus assuming each behavior included in either category have the same weight, regardless of the frequency and duration of individual behaviors.

Lastly, future implementation of this tool should focus on adopting the scale to be used on commercial swine farms to allow veterinarians, producers, and caretakers the ability to objectively quantify pain and provide pain relief when needed. Future work developing and disseminating current training programs (https://animalpain.org/en/home-en/-pain.com) to those working directly with pigs is needed.

**Table 9. Median (amplitude) of the UPAPS scores for castrated (n = 14) and non-painful female piglets (n = 6) awake piglets at M2.**

| Items | Female | Castrated |
|---|---|---|
| **Posture** | 0[b] (0–0) | 2[a] (0–3) |
| **Interaction and Interest** | 0[b] (0–0) | 2[a] (0–3) |
| **Activity** | 0[b] (0–2) | 3[a] (0–3) |
| **Attention to the affected area** | 0[b] (0–2) | 2[a] (0–3) |
| Elevates pelvic limb | 0 (0–1) | 0 (0–1) |
| Scratches the painful area | 0 (0–1) | 0 (0–1) |
| Moves after injury | 0[b] (0–1) | 0[a] (0–1) |
| Sits with difficulty | 0[b] (0–1) | 1[a] (0–1) |
| **Miscellaneous behavior** | 0.5[b] (0–2) | 2[a] (0–3) |
| Wags tail | 0[b] (0–1) | 1[a] (0–1) |
| Bites objects | 0 (0–0) | 0 (0–0) |
| Head is below the spinal column | 0 (0–1) | 1 (0–1) |
| Difficulty overcoming obstacles | 0 (0–1) | 0 (0–1) |
| **UPAPS** | 1[b] (0–6) | 9.5[a] (1–12) |

UPAPS—Unesp-Botucatu pig acute composite pain scale; Different letters express significant differences between moments where a > b, according to the mixed linear model [27]. M2: 15-min post-procedure, before rescue analgesia.

## Limitations

The current study had some limitations and constraints. The animals used to determine construct validity were non-painful female piglets. This limitation was driven by the small number of male piglets available to complete the study and, although unlikely, sexual differences in behavior may have influenced results. Behavioral differences are typically not present until pigs reach sexual maturity and the authors are unaware of other studies reporting a sex effect, as is demonstrated in other species such as the rat [55, 56]. Despite the small number of non-painful piglets enrolled, sample size requirements were calculated and met, and significance and validation of the tool was achieved. Of greater importance, male piglets served as their own control, thus the statistical tests analyzed utilized the true control of the male piglet prior to castration providing better insight into behavioral deviations presented after undergoing

**Table 10. Internal consistency of the UPAPS applied for castrated piglets awake (n = 14).**

| All items | Internal consistency | | | |
|---|---|---|---|---|
| | All castrated piglets (n = 39) | | Castrated piglets awake (n = 14) | |
| | Cronbach's α | McDonald's ω | Cronbach's α | McDonald's ω |
| | **0.85** | **0.88** | **0.89** | **0.92** |
| **Excluding each item below** | | | | |
| Posture | 0.80 | 0.85 | 0.86 | 0.90 |
| Interaction and interest | 0.83 | 0.88 | 0.87 | 0.91 |
| Activity | 0.82 | 0.87 | 0.86 | 0.90 |
| Attention to the affected area | 0.81 | 0.86 | 0.87 | 0.91 |
| Miscellaneous behaviors | 0.81 | 0.87 | 0.87 | 0.90 |

UPAPS: Unesp-Botucatu pig acute composite pain scale. Interpretation of the Cronbach's α coefficient values: 0.60–0.64 minimally acceptable; 0.65–0.69 acceptable; 0.70–0.74 good; 0.75–0.80 very good; > 0.80 excellent [42]. McDonald's omega interpretation: 0.70–0.84 is acceptable and > 0.85 is strong [41].

**Table 11. Specificity, sensitivity, and Youden index of optimal cut-off for UPAPS.**

| UPAPS | Specificity | | | Sensitivity | | | Youden Index | | |
|---|---|---|---|---|---|---|---|---|---|
| | Estimated | IC | | Estimated | IC | | Estimated | IC | |
| 0 | 0.00 | 0.00 | 0.00 | 1.00 | 1.00 | 1.00 | 0.00 | 0.00 | 0.00 |
| 1 | 0.57 | 0.39 | 0.75 | 1.00 | 1.00 | 1.00 | 0.57 | 0.39 | 0.75 |
| 2 | 0.71 | 0.54 | 0.86 | 0.93 | 0.82 | 1.00 | 0.64 | 0.36 | 0.86 |
| 3 | 0.75 | 0.57 | 0.89 | 0.93 | 0.82 | 1.00 | 0.68 | 0.39 | 0.89 |
| **4** | **0.79** | 0.61 | 0.93 | **0.93** | 0.82 | 1.00 | **0.71** | 0.43 | 0.93 |
| 5 | 0.79 | 0.61 | 0.93 | 0.79 | 0.64 | 0.93 | 0.57 | 0.25 | 0.86 |
| 6 | 0.86 | 0.71 | 0.96 | 0.71 | 0.57 | 0.86 | 0.57 | 0.29 | 0.82 |
| 7 | 0.89 | 0.79 | 1.00 | 0.68 | 0.50 | 0.82 | 0.57 | 0.29 | 0.82 |
| 8 | 0.96 | 0.89 | 1.00 | 0.68 | 0.50 | 0.82 | 0.64 | 0.39 | 0.82 |
| 9 | 0.96 | 0.89 | 1.00 | 0.54 | 0.36 | 0.71 | 0.50 | 0.25 | 0.71 |
| 10 | 0.96 | 0.89 | 1.00 | 0.50 | 0.32 | 0.68 | 0.46 | 0.21 | 0.68 |
| 11 | 1.00 | 1.00 | 1.00 | 0.36 | 0.18 | 0.54 | 0.36 | 0.18 | 0.54 |
| 12 | 1.00 | 1.00 | 1.00 | 0.21 | 0.07 | 0.39 | 0.21 | 0.07 | 0.39 |

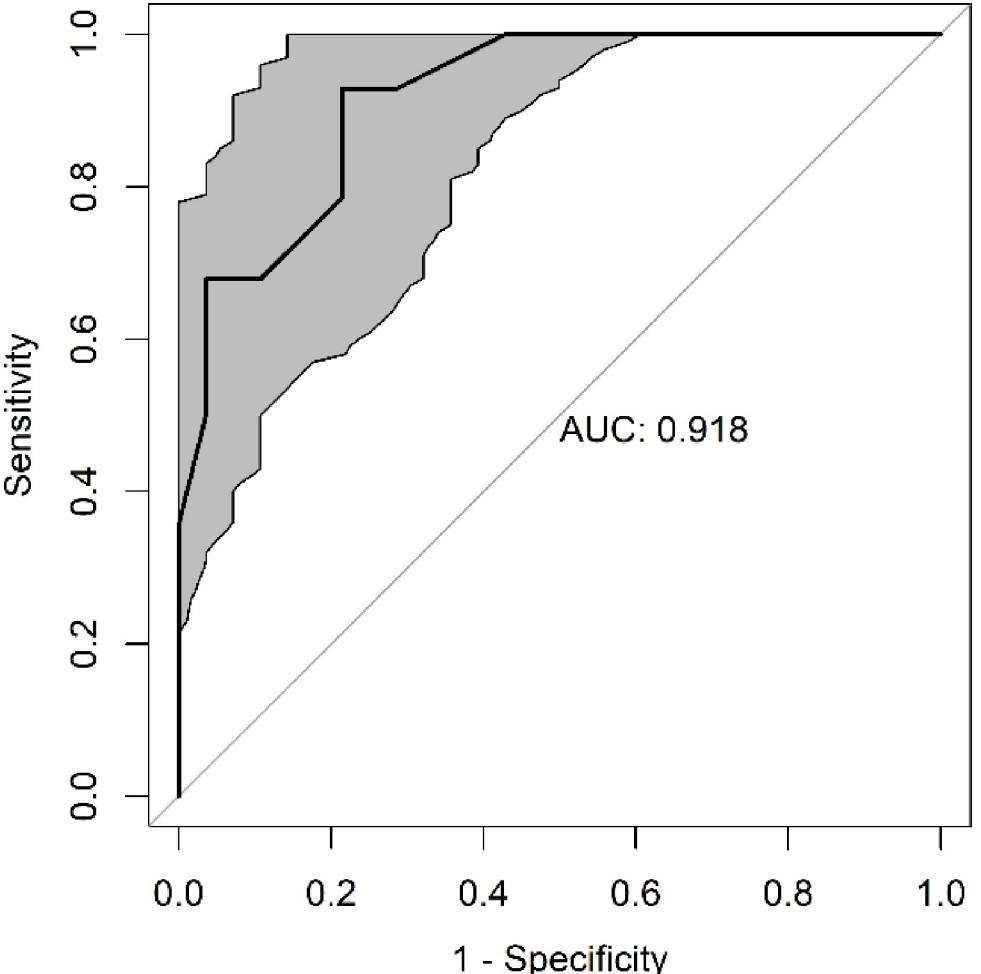

**Fig 9. Receiver Operating Characteristic (ROC) curve and area under the curve (AUC) of the UPAPS applied to castrated piglets awake (n = 14).** AUC values above 0.90 represent high discriminatory capacity (accuracy) of the scale [37]. The gray region represents the 95% confidence interval.

this painful procedure. Except for surgery, females underwent the same management and handling to simulate the same non-painful but potentially stressful conditions as males. The assessment of all castrated piglets 24h prior to castration served as a baseline for male piglet behavior prior to castration.

The validation of this tool at the research level and by remote assessment does not automatically guarantee that it may be applicable for live observations. Before UPAPS is deemed a valid instrument for use in both clinical and commercial settings devoid of remote assessment, this tool will need to be validated for live observation. Lastly, pre-weaned piglets spend significant periods of their time budget sleeping. Randomly selecting video clips can be challenging if piglets are asleep. Thus, future work must take into account how to randomly select video clips to assess piglet behavior, while maximizing opportunities to view active piglets.

## Conclusion

The UPAPS is a valid and reliable tool for remote assessment of post-castration pain in pre-weaned piglets. Further work with this tool is needed to validate and refine the scale for live observations on commercial swine farms utilizing caretakers and veterinarians responsible for animal care and management.

## Supporting information

**S1 Fig. Biplot of the principal component analysis of the UPAPS+nursing items applied to all castrated piglets (n = 39).** UPAPS–Unesp-Botucatu pig acute composite pain scale. Confidence ellipses indicate perioperative moments and pain scores. Moments: M1—preoperative; M2—postoperative, before rescue analgesia; M3—postoperative, after rescue analgesia; M4 - 24h after orchiectomy. Ellipses were constructed according to the assessment. Each circle corresponds to the score from one observer attributed to each piglet at each moment.
(TIF)

**S2 Fig. Violin plot of the total sum (median/amplitude) of the a) UPAPS+nursing (all castrated piglets; n = 39) and b) UPAPS+nursing (awake castrated piglets; n = 14) before and after orchiectomy in piglets.** The top and bottom box lines represent the interquartile range (25 to 75%), the line within the box represents the median, the extremes of the whiskers represent the minimum and maximum values, black lozenges (◆) represent the mean, black circles (•) represent outliers and width of the figures represent the distribution of data (wider sections represent a larger number of data). UPAPS: Unesp-Botucatu pig acute composite pain scale. Different letters express significant differences between moments where a > b > c, according to the generalized mixed linear models [24,26]. M1—preoperative; M2—postoperative before rescue analgesia.
(TIF)

**S1 Table. Loading values, eigenvalues and variance of the UPAPS+nursing and UPAPS items applied to all castrated piglets (n = 39) and only to awake castrated piglets (n = 14) based on principal components analysis.**
(TIF)

**S2 Table. Specificity and sensitivity and confidence interval (CI) of the 95% of the UPAPS applied for all castrated piglets (n = 39) and castrated piglets awake (n = 14).**
(TIF)

**S3 Table. Median (amplitude) of the UPAPS scores before and after orchiectomy in all castrated piglets (n = 39).**
(TIF)

**S4 Table. Median (amplitude) of the UPAPS scores before and after orchiectomy in castrated piglets awake (n = 14) during the assessment.**
(TIF)

**S5 Table. Median (amplitude) of the UPAPS scores for castrated (n = 14) and non-painful female piglets (n = 6) awake piglets, after orchiectomy during the assessment.**
(TIF)

**S6 Table. Item-total correlation and internal consistency of the UPAPS+nursing applied for all castrated piglets (n = 39).**
(TIF)

**S7 Table. Item-total correlation and internal consistency of the UPAPS+nursing applied for castrated piglets awake (n = 14).**
(TIF)

**S1 Dataset.**
(XLSX)

## Author Contributions

**Conceptualization:** M. D. Pairis- Garcia.

**Data curation:** I Robles, S. P. L. Luna, P. H. E. Trindade, A. V. Viscardi, E Tamminga, M. E. Lou, M. D. Pairis- Garcia.

**Formal analysis:** I Robles, S. P. L. Luna, P. H. E. Trindade.

**Funding acquisition:** V. R. Merenda.

**Investigation:** M. D. Pairis- Garcia.

**Methodology:** S. P. L. Luna, P. H. E. Trindade, M Lopez-Soriano, E Tamminga, M. E. Lou, M. D. Pairis- Garcia.

**Software:** P. H. E. Trindade.

**Supervision:** M. D. Pairis- Garcia.

**Validation:** S. P. L. Luna, P. H. E. Trindade.

**Visualization:** P. H. E. Trindade, M Lopez-Soriano, V. R. Merenda.

**Writing – review & editing:** S. P. L. Luna, M Lopez-Soriano, V. R. Merenda, A. V. Viscardi, E Tamminga, M. E. Lou, M. D. Pairis- Garcia.

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
