## [Decision Letter · Decision Letter 0]

24 Jun 2022

PONE-D-22-01216Validation of the UNESP-Botucatu pig composite acute pain scale (UPAPS) in piglets undergoing castration.PLOS ONE

Dear Dr. Pairis-Garcia,

Thank you for submitting your manuscript to PLOS ONE. After careful consideration, we feel that it has merit but does not fully meet PLOS ONE’s publication criteria as it currently stands. Therefore, we invite you to submit a revised version of the manuscript that addresses the points raised during the review process.

You will find the detailed reviewer comments below. As you will see, a number of them address methodological limitations, of which only some will be possible to address at this stage when the study has been completed. Please note that I expect that you address all comments. Your revised manuscript will be sent for review and the decision will depend on the outcome of that review.  

We look forward to receiving your revised manuscript.

Kind regards,

I Anna S Olsson, Ph.D.

Academic Editor

PLOS ONE

Journal Requirements:

Reviewers' comments:

Reviewer's Responses to Questions

**Comments to the Author**

1. Is the manuscript technically sound, and do the data support the conclusions?

Reviewer #1: Yes

Reviewer #2: No

2. Has the statistical analysis been performed appropriately and rigorously? 

Reviewer #1: Yes

Reviewer #2: Yes

3. Have the authors made all data underlying the findings in their manuscript fully available?

Reviewer #1: No

Reviewer #2: Yes

4. Is the manuscript presented in an intelligible fashion and written in standard English?

Reviewer #1: Yes

Reviewer #2: Yes

5. Review Comments to the Author

Reviewer #1: The authors describe the adaptation of a pain assessment scale (UPAPS) for new-born piglets.

They found the scale functioned well. While it was disappointing that the scale failed to detect the effects of rescue analgesia. However, this was most likely because the doses rate of flunixin was insufficient for pain control, so this does not detract from the study findings. The scale was found to have good to very good consistency both within and between scorers, so has excellent potential to be developed into a live assessment tool on commercial pig farms and therefore refine the welfare of many animals. The paper should be publishable once the authors have dealt with the following issue, some trivial, others more substantial. The queries below were constructed whilst reading the manuscript, so in some instances the authors have dealt with these at subsequent points in the manuscript. Nevertheless, they should amend the manuscript at the lines indicated to improve readability and comprehension. Note that although the authors state that all data have been made available, this is not technically correct. They say all relevant data are available in the manuscript and supporting information, but the supporting information contains results file, not original data.

Line 39: What does the adapted scale mean. Presumably this is described in the main text but should be define in the abstract if mentioned.

The abstract does not adequately describe what behaviours were measured. Sum scores is mentioned - what scores ? The purpose of an abstract is so a reader can quickly discover what the study is about without having to refer to the main text.

Line 66: Clarify here. The PGS has been shown to effective as indicated in citation 16. Do you mean it has not been validated in terms of being shown to reduce following drug treatment ?

Line 68: add 'the' before development

Line 69: insert 'the' before pain literature

Line 93: Did the stats control for litter?

Line 95: It’s nice to have example videos to look at but I am not sure how the piglets could be individually identified based on the marks that are visible. Are the example videos the ones that were scored ?

Line 104: What began the day they farrowed until weaning - access to water and feed? I don't think so.

Line 106: Personnel is plural and there was only one 'person'

Line 124: As females were used as controls - how were the assessors blinded to procedure? More importantly, as behaviour was recorded and scored before castration, the piglets served as their own controls, so why use any females at all?

Line 126: If so why does the piglet shown in this video have an ear tag and the others don’t?

https://www.youtube.com/watch?v=se70oYXcWFw

Line 133: Pigs were recorded not video.

Line 136: Why define the recording periods as M1, M2 etc when T1 and T2 would seem more logical? Better Still, to improve readability, why not use B (Baseline), PO (Post-op), PR (Post rescue) and 24h.

Line 138: Why only 4-minute clips when you have an hours’ worth of data. Seems a waste.

Line 144: As above - how did you blind between males and females. The scrotal sack could be visible in males.

Line 146: Please explain what is meant here. Do you mean because you only chose one 4 minute clip? Focal sampling would normally mean selecting random segments from the available footage.

Line 168: This is a repeat of an earlier statement and is not needed, just describe the adaptations not the reason for them.

174: Dos this mean only awake piglets were scored - it seems obvious it would be, therefore how did that affect numbers scored at each time-point?

Lines 171-180: This whole section is quite confusing. in some instances of sleeping, they were scored as 2, but for pain scoring they were scored 0 when sleeping.

Line 191: Why mention it in the abstract if it was excluded?

Line 200: Insert the relevant sums before and after castration

Line 207: Unusual to have citations in a table legend. Also, why use the term grouped moments for time points in the analysis. They are not moments after all but 4 min segments unless I missed something.

Line 213: You mean 'in' Table 1

Line 279: Should this not be the other way round in terms of what is bold or not.

Line 312: How moderate to intense?

Line 313: Therefore, the rescue analgesia protocol did not work ?

Line 348: Tables 7 and 8 are difficult to interpret. Add a description in the legends as to the meaning of one or 2 table items. What does the symbol d signify, for example when only a,b,c are mentioned.

Line 399: I would prefer the use of the term time(s) rather than moments throughout.

Line 414: Too much discussion devoted to why nursing was excluded.

Line 420: increased

Line 431: Insert ‘to’ after behaviour

Line 439: The fact that there is no chance of detecting pain in sleeping piglets is rather obvious. I am not sure why you bothered to include a sleeping version of the scale and test it so formally.

Line 449: Yes, so why choose the same 4-minute period from every video? A properly randomised scan sampling analysis would be more appropriate.

Line 451: Grammar: behavioural changes cannot n themselves be expressive. redraft this sentence

Line 453: change ‘was’ to were

Line 456: So in future what are you planning to do about it ?

Line 480: insert period punctuation before ‘training’

Line 486-489: Very long sentence

Line 497: Delete ‘the’ before human

Reviewer #2: The manuscript describes a thorough analysis of the validity of the UNESP-Botucatu pig composite acute pain scale (UPAPS). Although the statistical analysis is rather comprehensive, the conclusions are limited by major flaws in the experimental design (i.e. a small of group of female piglets as control). These limitations cannot be fixed by text revisions or changes in the statistical analysis, and may lead to unreliable conclusions regarding the pain scale, which could negatively impact the evaluation of piglet pain following castration. In addition, the method of castration (tearing of the spermatic cords) is painful and often contraindicated as compared to cutting of the cords. This choice is not justified nor explained in the manuscript, which poses an ethical problem.

If the manuscript is still considered for publication, the design limitations must be rightfully addressed, limitations clearly stated in the discussion, and conclusions nuanced. Further studies are required to validate this scale for piglet castration, and the conclusions should more clearly indicate so. Additional comments regarding specific sections of the manuscript include:

General:

- The whole text needs read-proofing and English check

- Very good use of illustration/video

Abstract:

- Some basic description of the pain scale (general elements recorded) is needed for comprehension

Introduction:

- ¨only 5% of the males piglets in some European countries are castrated with pain relief¨: it should be mentioned that this information is 6 years old. More recent statistics may be different – overall, pain relief is mandatory in many European countries, so this is a bit misleading.

Methodology:

- Power calculations are stated, but no justification is given for the lower number of control piglets included. This is a risk of self-replication. In addition, when considering only piglets awake, this sham groups falls to n=6, which is relatively low.

- Number of piglet per group analyzed is missing in Table 2, yet very important

- A description of the repartition of the tested piglets within litters is missing

- Litter effects are lacking as random effect in the statistical models (e.g. in the responsiveness and construct validity analysis)

- A complete and detailed ethical statement is missing. It is hard to understand why castration involved tearing of the spermatic cords, shown to be painful, and generally not recommended, rather than cutting. Incisions are also relatively large compared to standard practice. It is also not clear from the text why analgesia was not administered (if it is because pain was needed to induced for the sake of the study, or because the study aimed at resembling practice, this must be more clearly explained).

- It is unclear why only a 4 minute clip was used rather than a longer one. Considering that some of the behaviours are short-lasting or only observed at certain points in time (e.g. nursing), this choice can be limiting. Further justification is needed.

- Some items of the scale deserve a more detailed description, with a more precise choice of words. E.g. how is discomfort observed? Why is quietness part of the posture scoring, are vocalizations taken into consideration? Overall some of the categories are rather subjective, and deserve more explanation and critical discussion.

- The justification of why nursing was not retained in the analysis must not only be present in Table 1 but also in the materials & methods / results section.

- The section on distribution of scores could be stronger with some actual statistical testing, rather than descriptive statistics. These should include litter effects.

- In the multiple association section (PCA results), the ellipses are good graphical representations, but may not be enough to conclude on “on effect of pain”. Statistical models including PC result and moment (with litter as random effect) can support these claims in a more accurate way.

- It is unclear why specificity is only recorded in M1 and sensitivity only in M2. Why not look at these elements across the 4 recording time points? Further justification is needed.

Discussion:

- It is problematic that all shams were female. This is discussed a bit, but the discussion fails to consider potential cofounding effects (e.g. in terms of social interaction, see Exploration of early social behaviors and social styles in relation to individual characteristics in suckling piglets by Clouard et al)

- I miss a discussion on the scale in itself. Castration-related behaviours observed in past studies include e.g. huddling up or shivering. Why were these not included?

- The discussion on nursing behaviour as indicator of pain lacks a critical reflection on the methods. The fact that nursing was not relevant in this study may be due to the fact that it is not an appropriate indicator of pain, yes, but it can also be due to the study in itself: only 4 minutes clip, while nursing happens approx. 1 or 2 times per hour + nursing is a social behaviour that partly depends on the littermates.

- The discussion on using presence of certain behaviour rather than occurrence frequency deserves more attention. Looking at presence may be more effective, but it may also very much be less accurate (that may be why scratching did not increase in this study). This must be more clearly explained.

- In general, although results are promising, the discussion should be more critical regarding the method: larger-scale studies are still needed (including more piglets, across a wide age range), and the pain scale should also be validated externally by comparison to other quantitative methods to be able to conclude on sensitivity and reliability. Additional, there is lack of validation provided for the pain scale used in the older piglets.

6. PLOS authors have the option to publish the peer review history of their article (what does this mean?). If published, this will include your full peer review and any attached files.

Reviewer #1: No

Reviewer #2: No

---

## [Author Response · Author response to Decision Letter 0]

20 Aug 2022

Response to reviewers:

Reviewer #1: The authors describe the adaptation of a pain assessment scale (UPAPS) for new-born piglets.

They found the scale functioned well. While it was disappointing that, the scale failed to detect the effects of rescue analgesia. However, this was most likely because the doses rate of flunixin was insufficient for pain control, so this does not detract from the study findings. The scale was found to have good to very good consistency both within and between scorers, so has excellent potential to be developed into a live assessment tool on commercial pig farms and therefore refine the welfare of many animals. The paper should be publishable once the authors have dealt with the following issue, some trivial, others more substantial. The queries below were constructed whilst reading the manuscript, so in some instances the authors have dealt with these at subsequent points in the manuscript. Nevertheless, they should amend the manuscript at the lines indicated to improve readability and comprehension. Note that although the authors state that all data have been made available, this is not technically correct. They say all relevant data are available in the manuscript and supporting information, but the supporting information contains results file, not original data.

Response: Thank you for your time and effort in reviewing the manuscript. All comments have been reviewed and responded to accordingly

Line 39: What does the adapted scale mean. Presumably this is described in the main text but should be define in the abstract if mentioned.

The abstract does not adequately describe what behaviours were measured. Sum scores is mentioned - what scores ? The purpose of an abstract is so a reader can quickly discover what the study is about without having to refer to the main text.

Response: The abstract has been revised to clarify the adapted scale and include further description of the behavioral items measured

Line 66: Clarify here. The PGS has been shown to effective as indicated in citation 16. Do you mean it has not been validated in terms of being shown to reduce following drug treatment ?

Response: Sentence restructured to clarify PGS work

Line 68: add 'the' before development

Response: Added

Line 69: insert 'the' before pain literature

Response: Added

Line 93: Did the stats control for litter?

Response: Original statistical analysis did not include litter as an effect. Analysis was rerun to include litter as a random effect in the model for responsiveness to control for this effect in our sampling. The addition of litter as a random effect resulted in minimal changes that have been addressed in the results and Tables 7, 8, and 9.

Line 95: It’s nice to have example videos to look at but I am not sure how the piglets could be individually identified based on the marks that are visible. Are the example videos the ones that were scored?

Response: Example videos include both videos that were scored for this experiment and additional videos that were used for another project. Piglets were identified by either the markings on their back or the color ear tag. For example, posture 0 example is pointing at the pig with the black line on its back, followed by the piglet with the black open circle on the back and lastly the piglet with the purple ear tag. No markings were repeated within a litter and no color was repeated in a litter for pigs evaluated for behavior. 

Line 104: What began the day they farrowed until weaning - access to water and feed? I don't think so

Response: Statement revised for clarification

Line 106: Personnel is plural and there was only one 'person'

Response: Changed to person

Line 124: As females were used as controls - how were the assessors blinded to procedure? More importantly, as behaviour was recorded and scored before castration, the piglets served as their own controls, so why use any females at all?

Response: Thank you for the response. We agree that the use of the term “control” for the female piglets is misleading and has been re-written as follows: Female piglets served to assess the effect of sex, handling and social dynamics on the pain scale and were not considered true controls. In addition, video cameras were mounted 2.4m above the ground making it difficulty to visually observer piglet sex in video. However, there is still the possibility that sex was identified on camera and has now been clarified. 

Line 126: If so why does the piglet shown in this video have an ear tag and the others don’t?

Response: Please see previous response on videos. Videos with piglets with ear tags was collected from another study and was used solely as an example of the behavior. All piglets enrolled in this current study had markings located on their back for identification

Line 133: Pigs were recorded not video.

Response: Corrected

Line 136: Why define the recording periods as M1, M2 etc when T1 and T2 would seem more logical? Better Still, to improve readability, why not use B (Baseline), PO (Post-op), PR (Post rescue) and 24h.

Response: Thank you for your comments. I agree that the use of different acronyms may be more intuitive, recording periods have been named to maintain consistency with previous work on different species-pain scales to allow for more effective comparison of the work. The authors choose to keep the recording period names 

Line 138: Why only 4-minute clips when you have an hours’ worth of data. Seems a waste.

Response: 4-min clips were selected based on previous work conducted in pain scale research in livestock (Luna et al, 2020). Additionally, one objective of this work is to identify alternative methodologies to quantify pain sensitivity while minimizing labor, equipment and time requirements for observing behavior. Thus, the selection of 4-min was also conducted to evaluate if pig pain can be detected in shorter behavior observations timepoints

Line 144: As above - how did you blind between males and females. The scrotal sack could be visible in males.

Response: the authors agree and have included information stating the potential bias that may have occurred. However, camera angle and height made it very difficult to discern piglet sex so bias, although possible, would be limited.

Line 146: Please explain what is meant here. Do you mean because you only chose one 4 minute clip? Focal sampling would normally mean selecting random segments from the available footage.

Response: Agreed. Sentence has been re-written to emphasize that only one piglet was observed for each observation, not multiple piglets at one observation period. 

Line 168: This is a repeat of an earlier statement and is not needed, just describe the adaptations not the reason for them.

Response: Statement removed

174: Does this mean only awake piglets were scored - it seems obvious it would be, therefore how did that affect numbers scored at each time-point?

Response: Paragraph revised to clarify. Awake and asleep piglets were assessed. Piglet state was evaluated at the start of each observation. Piglets were categorized in one of two states: 1) Awake or 2) asleep. Piglet state was then used in the statistical model to to facilitate the discrimination between a piglet sleeping (classified as normal) compared to an awake piglet expressing normal non-painful behavior 

Lines 171-180: This whole section is quite confusing. in some instances of sleeping, they were scored as 2, but for pain scoring they were scored 0 when sleeping.

Response: Revised

Line 191: Why mention it in the abstract if it was excluded?

Response: Abstract and materials and method revised. Nursing behavior has traditionally been used as a common behavior to evaluate pain states in piglets. The authors felt it important to acknowledge nursing was evaluated and included in the assessment but is not considered a reliable behavior to assess as an item in the pain scale.

Line 200: Insert the relevant sums before and after castration

Response: The authors are unclear what the request is here and it appears number lines are not matching up. 

Line 207: Unusual to have citations in a table legend. Also, why use the term grouped moments for time points in the analysis. They are not moments after all but 4 min segments unless I missed something.

Response: Citations have been moved to superscripts of the table. The term “moment” has been used to maintain consistent terminology from previous published work on the pain scale.

Line 213: You mean 'in' Table 1

Response: Corrected

Line 279: Should this not be the other way round in terms of what is bold or not.

Response: Legend revised to provide better clarity 

Line 312: How moderate to intense?

Response: There is no current mathematical model to differentiate moderate to intense. For the purposes of this study we defined intense as pain expressed immediately post-castration (M2) and moderate as pain expressed 24 hour post-castration (M4). The manuscript has been updated to reflect this. 

Line 313: Therefore, the rescue analgesia protocol did not work?

Response: Correct. Sentence revised

Line 348: Tables 7 and 8 are difficult to interpret. Add a description in the legends as to the meaning of one or 2 table items. What does the symbol d signify, for example when only a,b,c are mentioned.

Response: Superscripts for all figures and tables have been revised

Line 399: I would prefer the use of the term time(s) rather than moments throughout.

Response: Thank you for your comments. I agree that the use of different acronyms may be more intuitive, recording periods have been named to maintain consistency with previous work on different species-pain scales to allow for more effective comparison of the work. The authors choose to keep the recording period names

Line 414: Too much discussion devoted to why nursing was excluded.

Response: Discussion section on nursing revised

Line 420: increased

Response: Discussion revised

Line 431: Insert ‘to’ after behaviour

Response: Discussion revised

Line 439: The fact that there is no chance of detecting pain in sleeping piglets is rather obvious. I am not sure why you bothered to include a sleeping version of the scale and test it so formally.

Response: Limitation section revised

Line 449: Yes, so why choose the same 4-minute period from every video? A properly randomised scan sampling analysis would be more appropriate.

Response: The authors agree and have provided further explanation on why randomly selected scan samples are difficult to conduct when piglet activity time periods are so short. This has been further added to the limitation section.

Line 451: Grammar: behavioural changes cannot n themselves be expressive. redraft this sentence

Line 453: change ‘was’ to were

Line 456: So in future what are you planning to do about it?

Line 480: insert period punctuation before ‘training’Line 486-489: Very long sentence

Line 497: Delete ‘the’ before human

Response: thank you for your thorough review of the discussion. We identify that the discussion was written in a way that was not clear and opportunities were missed to further explain some of the limitations with video selection and rescue analgesic. The discussion has been thoroughly revised, and we appreciate the guidance in making this manuscript more succinct

Reviewer #2: The manuscript describes a thorough analysis of the validity of the UNESP-Botucatu pig composite acute pain scale (UPAPS). Although the statistical analysis is rather comprehensive, the conclusions are limited by major flaws in the experimental design (i.e., a small of group of female piglets as control). These limitations cannot be fixed by text revisions or changes in the statistical analysis and may lead to unreliable conclusions regarding the pain scale, which could negatively impact the evaluation of piglet pain following castration. In addition, the method of castration (tearing of the spermatic cords) is painful and often contraindicated as compared to cutting of the cords. This choice is not justified nor explained in the manuscript, which poses an ethical problem.

If the manuscript is still considered for publication, the design limitations must be rightfully addressed, limitations clearly stated in the discussion, and conclusions nuanced. Further studies are required to validate this scale for piglet castration, and the conclusions should more clearly indicate so. Additional comments regarding specific sections of the manuscript include:

General:

- The whole text needs read-proofing and English check

- Very good use of illustration/video

Abstract:

- Some basic description of the pain scale (general elements recorded) is needed for comprehension

Response: Abstract revised

Introduction:

- ¨only 5% of the males piglets in some European countries are castrated with pain relief¨: it should be mentioned that this information is 6 years old. More recent statistics may be different – overall, pain relief is mandatory in many European countries, so this is a bit misleading.

Response: Statement clarified to prevent misleading. 

Methodology:

- Power calculations are stated, but no justification is given for the lower number of control piglets included. This is a risk of self-replication. In addition, when considering only piglets awake, this sham groups falls to n=6, which is relatively low

Response: Thank you for your comments. The term “control” was not an accurate description of the role of the female piglets enrolled on the study and has been changed throughout the manuscript. Given variation in pain sensitivity, the study was designed to permit male piglets to serve as their own control by collecting pain scale data prior to a painful event and collecting data during intense, moderate and mild pain states. The female non-painful piglets were enrolled on the study solely to take into account the effect of natural behavioral variation by day on pain scale results. Previous work published by two co-authors on this manuscript noted behavioral variations by day in castrated horses unassociated with painful states. This work conducted by Trindade and colleagues (2021) noted that pain-related behaviors of horses such as walking and looking out the window decreased, while behaviors such as resting, standing still, and resting pelvic limb increased during the night in comparison with morning or afternoon. Thus, naturally behaviors change throughout the day and providing a non-painful group of female piglets allowed us to take into account such variation in day and time effect on pain scale behavior. 

Trindade, P. H. E., Taffarel, M. O., & Luna, S. P. L. (2021). Spontaneous behaviors of post-orchiectomy pain in horses regardless of the effects of time of day, anesthesia, and analgesia. Animals, 11(6), 1629.

- Number of piglet per group analyzed is missing in Table 2, yet very important

Response: Piglet number per group has been included in the superscript of the table

- A description of the repartition of the tested piglets within litters is missing

- Litter effects are lacking as random effect in the statistical models (e.g., in the responsiveness and construct validity analysis)

Response: Original statistical analysis did not include litter as an effect. Analysis was rerun to include litter as a random effect in the model for responsiveness to control for this effect in our sampling. The addition of litter as a random effect resulted in minimal changes that have been addressed in the results and Tables 7, 8, and 9.

- A complete and detailed ethical statement is missing. It is hard to understand why castration involved tearing of the spermatic cords, shown to be painful, and generally not recommended, rather than cutting. Incisions are also relatively large compared to standard practice. 

Response: All male piglets located at this facility are required to undergo castration prior to weaning utilizing the approved farm Standard operating procedure and training protocols of the staff. 

It is also not clear from the text why analgesia was not administered (if it is because pain was needed to induce for the sake of the study, or because the study aimed at resembling practice, this must be more clearly explained).

Response: Methodology section revised to clarify lack of analgesic use

- It is unclear why only a 4-minute clip was used rather than a longer one. Considering that some of the behaviours are short-lasting or only observed at certain points in time (e.g. nursing), this choice can be limiting. Further justification is needed.

Response: 4-min clips were selected based on previous work conducted in pain scale research in livestock (Luna et al, 2020). Additionally, one objective of this work is to identify alternative methodologies to quantify pain sensitivity while minimizing labor, equipment, and time requirements for observing behavior. Thus, the selection of 4-min was also conducted to evaluate if pig pain can be detected in shorter behavior observations timepoints

- Some items of the scale deserve a more detailed description, with a more precise choice of words. E.g., how is discomfort observed? Why is quietness part of the posture scoring, are vocalizations taken into consideration? Overall, some of the categories are rather subjective, and deserve more explanation and critical discussion.

Response: Scale has been revised to include additional information on each behavioral item

- The justification of why nursing was not retained in the analysis must not only be present in Table 1 but also in the materials & methods / results section.

Response: Removal of nursing behavior is stated in results section 295-299.

- The section on distribution of scores could be stronger with some actual statistical testing, rather than descriptive statistics. These should include litter effects.

Response: Original statistical analysis did not include litter as an effect. Analysis was rerun to include litter as a random effect in the model for responsiveness to control for this effect in our sampling. The addition of litter as a random effect resulted in minimal changes that haven been addressed in the results and Tables 7, 8, and 9. The distribution of scores serves as a visual representation of the data for readers and is not intended to provide statistical meaning to the manuscript. The remaining statistical analyses presented in the paper serve that role. In addition, the authors added Horn’s parallel analysis and McDonald’s omega coefficient to the statistical package and these tests have been described further in the materials and methods. 

- In the multiple association section (PCA results), the ellipses are good graphical representations, but may not be enough to conclude on “on effect of pain”. Statistical models including PC result and moment (with litter as random effect) can support these claims in a more accurate way.

Response: Please see previous comments 

- It is unclear why specificity is only recorded in M1 and sensitivity only in M2. Why not look at these elements across the 4 recording time points? Further justification is needed.

Response: Specificity and sensitivity were only evaluated at two moments as these moments represented the only moments that were identified as 1) non-painful (M1) and 2) painful (M2) given the occurrence or no occurrence of the castration procedure. Calculating the specificity and sensitivity of the other moments is impossible given the pain states varies and intensity of pain is inconsistent. 

Discussion:

- It is problematic that all shams were female. This is discussed a bit, but the discussion fails to consider potential cofounding effects (e.g., in terms of social interaction, see Exploration of early social behaviors and social styles in relation to individual characteristics in suckling piglets by Clouard et al)

- I miss a discussion on the scale. Castration-related behaviours observed in past studies include e.g. huddling up or shivering. Why were these not included?

- The discussion on nursing behaviour as indicator of pain lacks a critical reflection on the methods. The fact that nursing was not relevant in this study may be due to the fact that it is not an appropriate indicator of pain, yes, but it can also be due to the study in itself: only 4 minutes clip, while nursing happens approx. 1 or 2 times per hour + nursing is a social behaviour that partly depends on the littermates.

- The discussion on using presence of certain behaviour rather than occurrence frequency deserves more attention. Looking at presence may be more effective, but it may also very much be less accurate (that may be why scratching did not increase in this study). This must be more clearly explained.

- In general, although results are promising, the discussion should be more critical regarding the method: larger-scale studies are still needed (including more piglets, across a wide age range), and the pain scale should also be validated externally by comparison to other quantitative methods to be able to conclude on sensitivity and reliability. Additional, there is lack of validation provided for the pain scale used in the older piglets.

Response: Thank you very much for the comments. We appreciate your constructive feedback and agree that there is opportunity for our team to revise the discussion to address the concerns you have noted. We have clarified the role of the non-painful female piglets. These piglets do not serve as the true control for the castrated piglets. Given individual variation in pain sensitivity, the experiment was designed to allow male piglets to serve as their own control, thus assessing and comparing pain scores prior to and following a painful event. The female non-painful piglets were enrolled on the study solely to take into account the effect of natural behavioral variation by day on pain scale results. Previous work published by two co-authors on this manuscript noted behavioral variations by day in castrated horses unassociated with painful states. This work conducted by Trindade and colleagues (2021) noted that pain-related behaviors of horses such as walking and looking out the window decreased, while behaviors such as resting, standing still, and resting pelvic limb increased during the night in comparison with morning or afternoon. Thus, naturally behaviors change throughout the day and providing a non-painful group of female piglets allowed us to take into account such variation in day and time effect on pain scale behavior.

---

## [Decision Letter · Decision Letter 1]

5 Dec 2022

 PONE-D-22-01216R1 Validation of the Unesp-Botucatu pig composite acute pain scale (UPAPS) in piglets undergoing castration. PLOS ONE

Dear Dr. Pairis-Garcia,

Thank you for submitting your manuscript to PLOS ONE. After careful consideration, we feel that it has merit but does not fully meet PLOS ONE’s publication criteria as it currently stands. Therefore, we invite you to submit a revised version of the manuscript that addresses the points raised during the review process. You find the reviewer comments below. Please submit your revised manuscript by Jan 18 2023 11:59PM. If you will need more time than this to complete your revisions, please reply to this message or contact the journal office at plosone@plos.org. Please include the following items when submitting your revised manuscript: A rebuttal letter that responds to each point raised by the academic editor and reviewer(s). You should upload this letter as a separate file labeled 'Response to Reviewers'.A marked-up copy of your manuscript that highlights changes made to the original version. You should upload this as a separate file labeled 'Revised Manuscript with Track Changes'.An unmarked version of your revised paper without tracked changes. You should upload this as a separate file labeled 'Manuscript'. If applicable, we recommend that you deposit your laboratory protocols in protocols.io to enhance the reproducibility of your results. Protocols.io assigns your protocol its own identifier (DOI) so that it can be cited independently in the future. For instructions see: https://journals.plos.org/plosone/s/submission-guidelines#loc-laboratory-protocols. Additionally, PLOS ONE offers an option for publishing peer-reviewed Lab Protocol articles, which describe protocols hosted on protocols.io. Read more information on sharing protocols at https://plos.org/protocols?utm_medium=editorial-email&utm_source=authorletters&utm_campaign=protocols.

We look forward to receiving your revised manuscript.

Kind regards,

I Anna S Olsson, Ph.D.

Academic Editor

PLOS ONE

Journal Requirements:

Reviewers' comments:

Reviewer's Responses to Questions

**Comments to the Author**

1. If the authors have adequately addressed your comments raised in a previous round of review and you feel that this manuscript is now acceptable for publication, you may indicate that here to bypass the “Comments to the Author” section, enter your conflict of interest statement in the “Confidential to Editor” section, and submit your "Accept" recommendation.

Reviewer #1: All comments have been addressed

Reviewer #2: (No Response)

2. Is the manuscript technically sound, and do the data support the conclusions?

Reviewer #1: Yes

Reviewer #2: Yes

3. Has the statistical analysis been performed appropriately and rigorously?

Reviewer #1: Yes

Reviewer #2: Yes

4. Have the authors made all data underlying the findings in their manuscript fully available?

Reviewer #1: Yes

Reviewer #2: No

5. Is the manuscript presented in an intelligible fashion and written in standard English?

Reviewer #1: Yes

Reviewer #2: Yes

6. Review Comments to the Author

Reviewer #1: In my opinion the authors have adequately addressed my queries, but I feel they have not fully replied to the comments of Reviewer 2, but I leave that up to reviewer 2 to decide.

Line 384: change decrease and increase to decreased and increased

Line 413: change 'The UPAPS' to 'the UPAPS'

Reviewer #2: I would like to congratulate the authors on a thorough work on improving the article.

The role of the female piglets and use of males as their own controls is now much clearer, and most of the comments on the statistical analysis and content of the article have been addressed.

I still, however, miss an important discussion regarding the content of the scale and the distribution of the behaviours. Although this aspect may have been addressed in previous work, it is central to the discussion, and should be rightfully acknowledged. In particular, I have doubts regarding the pertinence of using the presence of one or more ‘miscellaneous’ behaviours to reach a score for that category. This method implies that behaviours A, B, C or D have a comparable importance for pain recording, and does not take into consideration their intensity or repetition. For instance, prostration may be a stronger sign of pain than biting the bars or objects, and the duration of prostration (4 continuous minutes vs. 2 sec of the behaviour) may also be of importance. Yet these elements are not considered by the scale. The same goes for the ‘attention to the affected area’ category.

For more insights over the relative importance of different behaviours for pain recording in the context of piglet castration, I suggest to read the review ‘Optimal methods of documenting analgesic efficacy in neonatal piglets undergoing castration’ by Sheil and Polkinghorne.

I would like this discussion addressed, as it may affect the reliability and specificity of the scale. In that regard, the scale should be validated by quantitative data.

I am also missing, in the materials and method, an explanation of the distribution of males per litter. Was there a single tested male per litter? 2, 3, 4?

This element is of importance considering that you have social behaviours in the scale. If all males were castrated, and therefore potentially in pain or disturbed, and all females were handled, therefore potentially stressed, can you really say anything about the interaction of a specific individual with its littermates? Again, this warrants a discussion on the behaviours included in the scale.

Lastly, the dataset used to perform the analysis should be made available.

7. PLOS authors have the option to publish the peer review history of their article (what does this mean?). If published, this will include your full peer review and any attached files.

**Do you want your identity to be public for this peer review?** For information about this choice, including consent withdrawal, please see our Privacy Policy.

Reviewer #1: No

Reviewer #2: No

---

## [Author Response · Author response to Decision Letter 1]

14 Dec 2022

Reviewer #1: In my opinion the authors have adequately addressed my queries, but I feel they have not fully replied to the comments of Reviewer 2, but I leave that up to reviewer 2 to decide.

Line 384: change decrease and increase to decreased and increased

Response: Updated

Line 413: change 'The UPAPS' to 'the UPAPS'

Response: updated

Reviewer #2: I would like to congratulate the authors on a thorough work on improving the article.

The role of the female piglets and use of males as their own controls is now much clearer, and most of the comments on the statistical analysis and content of the article have been addressed.

I still, however, miss an important discussion regarding the content of the scale and the distribution of the behaviours. Although this aspect may have been addressed in previous work, it is central to the discussion, and should be rightfully acknowledged. In particular, I have doubts regarding the pertinence of using the presence of one or more ‘miscellaneous’ behaviours to reach a score for that category. This method implies that behaviours A, B, C or D have a comparable importance for pain recording, and does not take into consideration their intensity or repetition. For instance, prostration may be a stronger sign of pain than biting the bars or objects, and the duration of prostration (4 continuous minutes vs. 2 sec of the behaviour) may also be of importance. Yet these elements are not considered by the scale. The same goes for the ‘attention to the affected area’ category.

For more insights over the relative importance of different behaviours for pain recording in the context of piglet castration, I suggest to read the review ‘Optimal methods of documenting analgesic efficacy in neonatal piglets undergoing castration’ by Sheil and Polkinghorne.

I would like this discussion addressed, as it may affect the reliability and specificity of the scale. In that regard, the scale should be validated by quantitative data.

This element is of importance considering that you have social behaviours in the scale. If all males were castrated, and therefore potentially in pain or disturbed, and all females were handled, therefore potentially stressed, can you really say anything about the interaction of a specific individual with its littermates? Again, this warrants a discussion on the behaviours included in the scale.

Response: Thank you for your insight and reference to the article. We have updated the discussion to include limitations to the pain scale and address the impact of social behavior and behavior frequency and appreciate your constructive feedback to improve the discussion.

I am also missing, in the materials and method, an explanation of the distribution of males per litter. Was there a single tested male per litter? 2, 3, 4?

Response: information added in materials and methods

Lastly, the dataset used to perform the analysis should be made available.

Response: Included

---

## [Decision Letter · Decision Letter 2]

15 Mar 2023

PONE-D-22-01216R2

Validation of the Unesp-Botucatu pig composite acute pain scale (UPAPS) in piglets undergoing castration.

PLOS ONE

Dear Dr. Pairis-Garcia,

Thank you for submitting your manuscript to PLOS ONE. After careful consideration, we feel that it has merit but does not fully meet PLOS ONE’s publication criteria as it currently stands. Therefore, we invite you to submit a revised version of the manuscript that addresses the points raised during the review process.

Whereas the issues raised by the reviewers have been addressed, there are a few remaining minor issues.

Minor language issues:

line 57 instead of "pharmaceutical therapies" say "pharmaceutical pain control" as this is what is important in the context

line 99 change to "All male pigs in each litter were enrolled in the study"

line 149 "and one, four-minute video clips" is not understandable - something missing?

line 117 by saying that the sows had ad libitum access to one nipple and one feeder, you say they have access to the devices but you don't actually say anything about their access to water and feed!

line 120 "one-trained" should be "one trained"

line 122 please be specific as to whether the piglets were suspended by their hindlegs or placed with horizontal support

line 163 vs line 187 and any other place in the manuscript where you refer to specific tasks carried out by specific persons: be consistent in how you refer to this and to their identity. Ideally something like "by two observers (initialsA, initialsB)" or for 187 "by the same person (PHET)"

line 346 change to "is a critical welfare issue and pain negatively impacts"

line 363 "is often difficult to capture on video" - it doesn't seem to be the video that is the issue but the short time frame that you are using for the analysis, as you identify on lines 365-366. Please revise accordingly.

line 421 "subtly", not "subtlety"

line 425-26 Please change to "Male piglet number per litter in this study varied".

Apologies for the delay.

We look forward to receiving your revised manuscript.

Kind regards,

I Anna S Olsson, Ph.D.

Academic Editor

PLOS ONE

Journal Requirements:

Reviewers' comments:

Reviewer's Responses to Questions

**Comments to the Author**

1. If the authors have adequately addressed your comments raised in a previous round of review and you feel that this manuscript is now acceptable for publication, you may indicate that here to bypass the “Comments to the Author” section, enter your conflict of interest statement in the “Confidential to Editor” section, and submit your "Accept" recommendation.

Reviewer #2: All comments have been addressed

 2. Is the manuscript technically sound, and do the data support the conclusions?

Reviewer #2: Yes

 3. Has the statistical analysis been performed appropriately and rigorously?

Reviewer #2: Yes

 4. Have the authors made all data underlying the findings in their manuscript fully available?

Reviewer #2: Yes

 5. Is the manuscript presented in an intelligible fashion and written in standard English?

Reviewer #2: Yes

 6. Review Comments to the Author

Reviewer #2: The authors have adapted the discussion, and adequately addressed the limits of the scale. The paper can, in my view, be published.

 7. PLOS authors have the option to publish the peer review history of their article (what does this mean?). If published, this will include your full peer review and any attached files.

Reviewer #2: No

---

## [Author Response · Author response to Decision Letter 2]

17 Mar 2023

Minor language issues:

line 57 instead of "pharmaceutical therapies" say "pharmaceutical pain control" as this is what is important in the context

Response: changed

line 99 change to "All male pigs in each litter were enrolled in the study"

Response: added

line 149 "and one, four-minute video clips" is not understandable - something missing?

Response: revised

line 117 by saying that the sows had ad libitum access to one nipple and one feeder, you say they have access to the devices but you don't actually say anything about their access to water and feed!

Response: revised

line 120 "one-trained" should be "one trained"

Response: changed

line 122 please be specific as to whether the piglets were suspended by their hindlegs or placed with horizontal support

Response: revised

line 163 vs line 187 and any other place in the manuscript where you refer to specific tasks carried out by specific persons: be consistent in how you refer to this and to their identity. Ideally something like "by two observers (initialsA, initialsB)" or for 187 "by the same person (PHET)"

Response: Added

line 346 change to "is a critical welfare issue and pain negatively impacts"

Response: revised

line 363 "is often difficult to capture on video" - it doesn't seem to be the video that is the issue but the short time frame that you are using for the analysis, as you identify on lines 365-366. Please revise accordingly.

Response: revised

line 421 "subtly", not "subtlety"

Response: Changed

line 425-26 Please change to "Male piglet number per litter in this study varied".

Response: changed

---

## [Editor Report · Decision Letter 3]

27 Mar 2023

Validation of the Unesp-Botucatu pig composite acute pain scale (UPAPS) in piglets undergoing castration.

PONE-D-22-01216R3

Dear Dr. Pairis-Garcia,

We’re pleased to inform you that your manuscript has been judged scientifically suitable for publication and will be formally accepted for publication once it meets all outstanding technical requirements.

Kind regards,

I Anna S Olsson, Ph.D.

Academic Editor

PLOS ONE
---

## [Editor Report · Acceptance letter]

4 Apr 2023

PONE-D-22-01216R3 

Validation of the Unesp-Botucatu pig composite acute pain scale (UPAPS) in piglets undergoing castration. 

Dear Dr. Pairis- Garcia:

I'm pleased to inform you that your manuscript has been deemed suitable for publication in PLOS ONE. Congratulations! Your manuscript is now with our production department. 

Kind regards, 

on behalf of

Dr. I Anna S Olsson 

Academic Editor

PLOS ONE